# Self-Supervised Discovery of Neural Circuits in Spatially Patterned Neural Responses with Graph Neural Networks

**Kijung Yoon**
Department of Electronic Engineering
Department of Artificial Intelligence
Hanyang University
Seoul, Korea 04763
kiyoon@hanyang.ac.kr

## Abstract

Inferring synaptic connectivity from neural population activity is a fundamental challenge in computational neuroscience, complicated by partial observability and mismatches between inference models and true circuit dynamics. In this study, we propose a graph-based neural inference model that simultaneously predicts neural activity and infers latent connectivity by modeling neurons as interacting nodes in a graph. The architecture features two distinct modules: one for learning structural connectivity and another for predicting future spiking activity via a graph neural network (GNN). Our model accommodates unobserved neurons through auxiliary nodes, allowing for inference in partially observed circuits. We evaluate this approach using synthetic data generated from ring attractor network models and real spike recordings from head direction cells in mice. Across a wide range of conditions, including varying recurrent connectivity, external inputs, and incomplete observations, our model reliably resolves spurious correlations and recovers accurate weight profiles. When applied to real data, the inferred connectivity aligns with theoretical predictions of continuous attractor models. These results highlight the potential of GNN-based models to infer latent neural circuitry through self-supervised structure learning, while leveraging the spike prediction task to flexibly link connectivity and dynamics across both simulated and biological neural systems.

## 1 Introduction

Understanding how neural circuits compute and adapt requires identifying the strength of synaptic transmission between neurons, as this knowledge reveals how the structure of a circuit shapes its computational properties. Advances in recording simultaneous activity from large populations of neurons have driven significant interest in using statistical methods [1–7] to infer interactions and estimate connectivity between neurons across entire circuits. Despite this progress, statistical inference methods face two primary challenges: first, no recording technique can capture the activity of all neurons within a circuit, and second, the inference models may fail to accurately represent the underlying generative dynamical system. As a result, these limitations can lead to substantial differences between inferred and true connectivity.

The discrepancy in connectivity inference primarily originates from instances in which weakly connected or unconnected neurons exhibit strong activity correlations. This is a characteristic feature of strongly recurrent networks, which maintain persistent memory states through the principle of pattern formation [8–10]—a process where simple, spatially localized competitive interactions produce stable spatial activity patterns. To address these challenges, we design continuous attractor networks capable of generating spatially patterned neural responses [11–15] and focus on evaluating

how effectively a circuit inference model can explain away the correlations that arise from co-activated neurons.

To this end, we propose the use of graph neural networks (GNNs), a robust framework for modeling complex dynamics in physical systems with numerous interacting entities, such as particles or atoms [16–19]. In our approach, neurons are represented as nodes and their connections as edges within a GNN-based architecture. The model incorporates two functionally distinct modules: one designed to learn structural connectivity across the network and another dedicated to predicting spiking activity based on simultaneous, circuit-wide neural recordings. Instead of relying on supervised learning, we aim to extract network connectivity in a self-supervised manner by training the model to predict subsequent spike events in neural populations, with the model's latent representation serving to describe the inferred connectivity. An additional feature of our model is its ability to account for unobserved neurons by adding extra nodes to the graph. Under a transductive framework [20, 21], the connectivity among observed neurons is inferred, allowing message passing to occur throughout the entire neural circuit, including both observed and hidden neurons. This formulation implicitly leverages hidden neurons to explore the influence of unobserved components in circuit inference.

We perform extensive experiments on neural spike data generated from highly structured ring networks under various conditions to systematically evaluate the inference performance of the proposed framework. Our analysis begins with a fully observed network without external input drives, enabling us to isolate challenges arising from mismatches between the generative system and the inference model. In this setting, we demonstrate that the proposed approach resolves correlations from unconnected neurons at least 70% more effectively than advanced statistical inference methods. Furthermore, we show that the improved quality of circuit inference consistently holds across diverse configurations of the generative model, including stimulus-driven conditions, different recurrent weight profiles, fully versus partially observed networks, and extends to real neural recordings from behaving mice.

The paper is structured as follows. Section 2 provides an overview of related work. Section 3 introduces the recurrent network models used to generate neural spike data and details the proposed GNN-based inference framework. Section 4 presents the experimental results, while Section 5 concludes the study.

## 2   Related Work

Estimating network connectivity from large population recordings has been a long-standing challenge in computational neuroscience. One prominent line of research focuses on probabilistic modeling techniques, including maximum entropy-based inverse Ising models [22, 2, 23] and minimum probability flow (MPF) [24, 25]. Both approaches leverage the Ising model to capture pairwise interactions among binary variables, such as neuronal spiking activity, to reconstruct functional connectivity graphs. Maximum entropy models ensure interpretability by maximizing likelihood under empirical constraints (e.g., correlations) but require computationally expensive estimation of the partition function. In contrast, MPF addresses this limitation by minimizing the probability flow between observed and unobserved states, bypassing the partition function. This makes MPF more scalable and computationally efficient for inferring connectivity in large neural networks.

Another widely used method involves $\ell_1$-regularized logistic regression to promote sparsity in connectivity estimates by penalizing the number of nonzero parameters [26, 27]. In this framework, $\ell_1$-regularized logistic regression is performed for each variable against all others, with the sparsity pattern of the regression coefficients used to infer the network's neighborhood structure. This technique is particularly effective for high-dimensional Ising model selection, supporting connectivity inference for large scale datasets with complex correlations. A related framework employs the use of generalized linear models (GLMs), originally formalized by Nelder and Wedderburn [28], to relate a linear predictor to an output variable through a link function. GLMs have been extensively applied to model spatio-temporal interactions and stimulus dependencies [29, 30, 1], predicting circuit activity by associating observed spiking activity with intrinsic factors such as spike history and external covariates like stimuli or movement. By explicitly modeling the influence of neurons on one another, these likelihood-based approaches treat network connectivity as parameters to be learned, reducing spurious interactions in network inference.

Model-based approaches have demonstrated significant success in characterizing neural interactions and dependencies, especially within sensory systems. However, even with extensive neuronal activity recordings, these models have been reported to inaccurately estimate effective connectivity in memory-

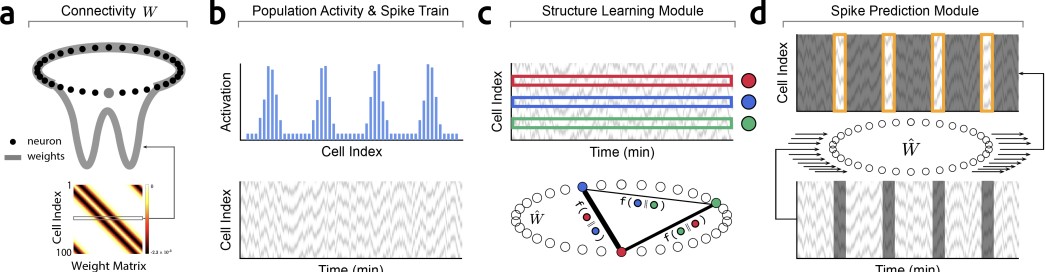

Figure 1: Overview of the generative and inference network model. (a) A ring network with a Mexican-hat connectivity profile, where each neuron is connected to others with the same weight pattern. The full weight matrix is shown below. (b) Simulated network activity, including synaptic activation (top) and spike raster plot across neurons over time (bottom). (c) Structure learning module that estimates synaptic connection strengths based on pairwise spike activity. (d) Spike prediction module that leverages the inferred connectivity to predict future spike times from past neural activity.

related (i.e., strongly recurrent) networks compared to sensory-driven circuits [31]. Consequently, the extent to which inferred connectivity faithfully reflects biological neuronal connections remains an open question—one that we seek to explore in this study.

## 3  Method

In this section, we introduce strongly recurrent networks that produce spatially structured activity patterns for generating neural spike data, outline the proposed GNN-based network inference approach[1], and detail key modifications of the inference model across different generative model configurations.

### 3.1  Generative recurrent network models

We simulate structured neural activity using a ring network of $N$ neurons with recurrent connectivity defined by a local Mexican-hat profile (Figure 1a). This architecture supports the formation of stable, spatially periodic activity patterns under a uniform excitatory drive (Figure 1b). Two spike generation models are considered: a threshold-crossing model and a linear-nonlinear Poisson (LNP) model. Both models integrate recurrent input and a shared feed-forward drive, but differ in their spike emission mechanisms, with the former using deterministic thresholding and the latter using stochastic Poisson sampling. These differences lead to distinct spike train statistics. We fix parameters such that multiple co-active activity bumps emerge, providing a challenging testbed for connectivity inference. Full equations and parameter settings are described in Appendix A.

### 3.2  Inference network models

We collect spike data from the generative network models over an 8-minute period with a time step of $\Delta t = 0.1$ ms, representing the spike trains as $\mathbf{x} \in \{0, 1\}^{N \times L}$, where $N$ denotes the number of neurons and $L$ corresponds to the number of time steps (Figure 1b, bottom). The recorded spike train is then processed by a structure learning module to infer the underlying neural circuitry, followed by a spike prediction module that concurrently predicts the activity of multiple neurons.

#### 3.2.1  Structure learning module

The objective of this module is to estimate the pairwise connection strength $w_{ij}$ for every pair $(\mathbf{x}_i, \mathbf{x}_j)$, where $\mathbf{x}_i = (\mathbf{x}_i^1, \ldots, \mathbf{x}_i^L)$ represents the spike train of $i$-th neuron. To achieve this, we apply a 1D convolution $f_{\text{Conv1D}}$ with 32 kernels, each having a size of $2\tau/\Delta t$, which is twice the synaptic time constant $\tau$ of the generative model. The convolution uses a stride equal to 20% of $\tau$ across each input spike train. The resulting feature maps are vectorized along the time dimension and passed through a fully connected layer $f_{\text{out}}$ to produce a reduced-dimensional output embedding vector $\mathbf{z}_i$ (Figure 1c, top):

$$\mathbf{z}_i = f_{\text{out}}\left(\text{vec}\left(f_{\text{Conv1D}}(\mathbf{x}_i)\right)\right) \qquad (1)$$

We include batch normalization [33] immediately after the ReLU activation function to improve training stability. Next, we concatenate the embeddings $(\mathbf{z}_i, \mathbf{z}_j)$ for every neuron pair and input them

---

[1]A preliminary version of this work appeared in *NeurIPS 2022 Workshop on Symmetry and Geometry in Neural Representations* [32].

into a multilayer perceptron (MLP) with two hidden layers of 32 units each to estimate the network connectivity strength (Figure 1c, bottom):

$$w_{ij} = \text{MLP}\left([\mathbf{z}_i, \mathbf{z}_j]\right) \tag{2}$$

The inferred weight matrix $\mathbf{w} \in \mathbb{R}^{N \times N}$, containing all pairwise coupling strengths $w_{ij}$ between neurons $i$ and $j$, is assumed to be symmetric and free of self-connections. However, we do not impose rotation invariance on $\mathbf{w}$, a constraint that governs the target recurrent weight strengths in the generative models. In other words, neurons in the proposed inference model are not required to share identical outgoing synaptic weights with all other neurons within the ring network.

### 3.2.2 Spike prediction module

This module aims to predict the future activity of the generative network by modeling the dynamics of interacting neurons, expressed as $p_\theta\left(\mathbf{x}^{t+1}|\mathbf{x}^t, \ldots, \mathbf{x}^1, \mathbf{w}\right)$, given the latent circuitry $\mathbf{w}$ and the past spike history. Specifically, this is done by modeling the sequential probability distribution of spike trains over time:

$$p(\mathbf{x}|\mathbf{w}) = \prod_{t=1}^{T} p\left(\mathbf{x}^{t+1}|\mathbf{x}^t, \ldots, \mathbf{x}^1, \mathbf{w}\right) \tag{3}$$

where $\mathbf{x}^t = (\mathbf{x}_1^t, \ldots, \mathbf{x}_N^t)$ represents the spike activity of all $N$ neurons at time $t$. This approach not only accounts for the temporal dependencies in neural activity but also integrates the connectivity structure learned by the structure learning module. To this end, we employ a GNN message-passing operation:

$$\mathbf{h}_i^t = f_{\text{enc}}\left(\mathbf{x}_i^{t-\ell+1:t}\right) \tag{4}$$

$$\mathbf{m}_{ij}^t = \phi\left([\mathbf{h}_i^t, \mathbf{h}_j^t]\right) \tag{5}$$

$$\log(\lambda_i^{t+1}) = f_{\text{dec}}\left(\mathbf{h}_i^{t+1}\right) \tag{7}$$

$$\mathbf{h}_i^{t+1} = \psi\left(\sum_{j \in \mathcal{N}(i)} w_{ij} \cdot \mathbf{m}_{ij}^t, \mathbf{h}_i^t\right) \tag{6}$$

$$p\left(\mathbf{x}^{t+1}|\mathbf{x}^{t-\ell+1:t}, \mathbf{w}\right) = \text{Pois}(\boldsymbol{\lambda}^{t+1}) \tag{8}$$

The first expression computes the initial embedding $\mathbf{h}_i^t$ for neuron $i$ by encoding its spike history over the past $\ell \,(= 2\tau/\Delta t)$ time steps using the encoder $f_{\text{enc}}(\cdot)$, which is functionally equivalent to the 1D convolution in Eq.(1) with shared parameters. The message vector $\mathbf{m}_{ij}^t$ representing the information transmitted from neuron $j$ to neuron $i$ is then computed by applying the function $\phi$ to the concatenated current states of both neurons. Afterwards, the neuronal state $\mathbf{h}_i^{t+1}$ is updated from $\mathbf{h}_i^t$ by aggregating incoming messages from neighboring neurons $j \in \mathcal{N}(i)$, where each message is weighted by the corresponding connectivity strength $w_{ij}$, the $(i,j)$ entry of $\mathbf{w}$, before being processed through the recurrent network $\psi$.

Finally, the decoding function $f_{\text{dec}}(\cdot)$ maps the neuronal states to the log firing rates, $\log \lambda^{t+1}$, which in turn determine the rates of an inhomogeneous Poisson process responsible for generating spikes at time $t+1$. Although the Poisson likelihood is formally conditioned only on a fixed-length window $\mathbf{x}^{t-\ell+1:t}$ and $\mathbf{w}$, the hidden state of a gated recurrent unit $\psi$ encodes a summary of all past activity up to time $t$. This is achieved through gated mechanisms that integrate temporal information and selectively retain or update relevant features from the spike history, enabling the model to capture full temporal dependencies.

### 3.2.3 Extensions for external inputs and hidden neurons

To better capture the complexities of biological neural systems, we extend our spike prediction framework in two key ways. First, we incorporate stimulus-driven embeddings alongside spike history to account for external inputs, enabling alignment with circular variables such as head direction. Second, to handle partially observed networks, we introduce hidden neurons whose embeddings are initialized via interpolation from nearby observed units, permitting full-graph message passing during inference. Full implementation details are provided in Appendix B.

## 4 Experiments

**Baselines** For our connectivity inference experiments, we benchmark our GNN-based model against three widely used baselines: GLM [1], sequential non-negative matrix factorization (seqNMF) [34], and tensor component analysis (TCA) [35] (see Appendix C for details). While the GLM stands

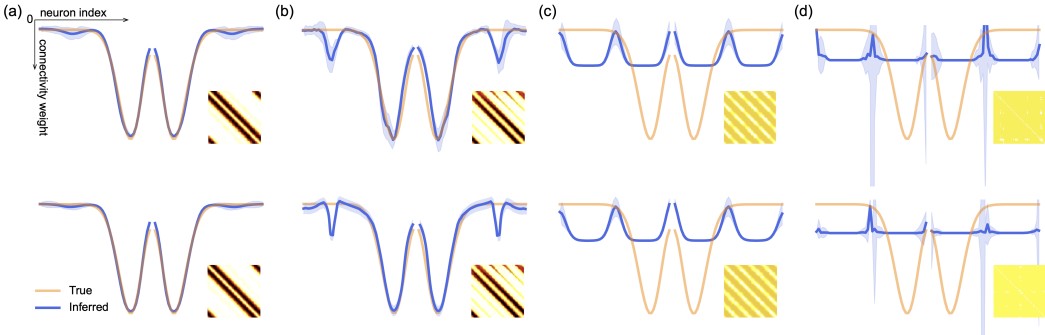

Figure 2: Quality of connectivity inference from spike train data generated by a fully observed network of 100 neurons. (a) Comparison of the ground-truth (orange) and inferred (blue) weight profiles obtained by the GNN-based inference model. The solid blue line represents the average inferred weights across three trials, each initialized with a different random seed, with the shaded blue region indicating $\pm 1$ standard deviation. The inset at the bottom right shows the full inferred weight matrix $\hat{\mathbf{W}}$. Each row corresponds to inference results from spike data simulated using the threshold-crossing model (top) and the LNP model (bottom). (b-d) Subsequent columns present the inference quality of baseline methods: (b) GLM, (c) seqNMF, and (d) TCA.

out for its ability to predict neural activity, particularly in sensory systems, it also effectively infers coupling effects among neurons. A key distinction between our framework and the GLM lies in how connectivity is represented and learned; we provide a detailed discussion of this difference in Appendix D. In contrast, TCA and seqNMF are not specifically designed for connectivity inference. Instead, they primarily aim to extract low-dimensional representations and capture neural dynamics. Nonetheless, we use them to identify low-dimensional neuron factors and examine their correlation structures, which can serve as a rough proxy for network connectivity.

**Datasets** We use both synthetic and real datasets within a unified simulation and evaluation framework. For synthetic data, we generate spiking activity from a 100-neuron ring network described in Section 3.1, simulating 8 minutes of activity at 0.1 ms resolution (4.8 million time steps). This setup is applied consistently across all synthetic experiments, including those with and without external inputs, varying recurrent connectivity structures, and under full or partial observability. For real data, we use publicly available HD recordings and motion tracking data from freely moving mice in an open-field environment [36]. The HD trajectories are treated as external inputs, while spike trains from HD cells are used to evaluate inference performance within the same framework. All datasets are partitioned into 80% training, 10% validation, and 10% testing splits. See Appendix E for details on preprocessing and integration of real data.

**Training** In our model, the optimization is framed as minimizing the Poisson negative log-likelihood, where the firing rate $\lambda_i^t$ governs the likelihood of observed spike activity $\mathbf{x}_i^t$. Since $\mathbf{w}$ is not a free parameter but a latent representation derived from the observed neural activity via deterministic transformations in the structure learning module, the optimization is performed solely over the model parameters $\Theta$. Given this dependence, $\mathbf{w}$ is not explicitly optimized but is instead updated indirectly as $\Theta$ is optimized. The final objective function for training is given by:

$$\Theta^* = \arg\min_{\Theta} \sum_{i=1}^{N} \sum_{t=1}^{T} \left( \lambda_i^t - \mathbf{x}_i^t \log \lambda_i^t \right) \tag{9}$$

We train the model using the Adam optimizer with a learning rate of $5 \times 10^{-4}$, and further make use of an exponential decay schedule in the learning rate.

**Metrics** To evaluate connectivity inference, we align weight vectors across neurons to a common phase and normalize for global scaling before computing the normalized inference error, $\Delta$, relative to ground truth. This accounts for rotational symmetry and scale ambiguity inherent in the network structure. In parallel, we assess spike prediction performance using a log-likelihood-based metric, $\mathcal{L}_{\text{bps}}$, that quantifies improvement over a homogeneous Poisson model, yielding an interpretable score in bits per spike. Full derivations, alignment procedures, and implementation details are provided in Appendix F.

Table 1: Performance metrics for connectivity inference and spike prediction in a fully observed network without external inputs. Results are averaged over the three trials shown in Figure 2.

|  | Thresh. | | LNP | |
| --- | --- | --- | --- | --- |
|  | $\Delta \downarrow$ | $\mathcal{L}_{\text{bps}} \uparrow$ | $\Delta \downarrow$ | $\mathcal{L}_{\text{bps}} \uparrow$ |
| GNN | **0.061** | **0.882** | **0.049** | **0.876** |
| GLM | 0.244 | 0.695 | 0.238 | 0.712 |
| seqNMF | 0.789 | – | 0.796 | – |
| TCA | 0.762 | – | 0.761 | – |

Table 2: Evaluation of inference and prediction accuracy for a fully observed network under external input conditions. Each value is the average over three independent trials.

|  | Thresh. | | LNP | |
| --- | --- | --- | --- | --- |
|  | $\Delta \downarrow$ | $\mathcal{L}_{\text{bps}} \uparrow$ | $\Delta \downarrow$ | $\mathcal{L}_{\text{bps}} \uparrow$ |
| GNN | **0.073** | **0.916** | **0.058** | **0.924** |
| GLM | 0.259 | 0.724 | 0.245 | 0.748 |
| seqNMF | 0.791 | – | 0.794 | – |
| TCA | 0.760 | – | 0.762 | – |

## 4.1 Fully Observed Network

We begin by assessing the accuracy of connectivity inference when the spiking activity of all neurons in the generative network is fully observed. A key finding is that all baseline methods tend to mistakenly infer connections to neurons that are either unconnected or only weakly connected. This is evident from the side dips in the weight profiles and the presence of multiple off-diagonal stripes in the inferred weight matrices (Figures 2b–2d). Such systematic inference errors stem from overestimated connections driven by strong correlations in neural activity [31], which arise from the global activity patterns intrinsic to our recurrent generative networks (see Figure 1b). Among the baselines, the GLM achieves the closest match to the true connectivity but still struggles to properly explain away these spurious correlations. In contrast, our proposed GNN inference model effectively suppresses these artifacts (Figure 2a), resulting in significantly lower inference errors (Table 1). This improvement is accompanied by superior spike prediction accuracy (Table 1), suggesting that the GNN-based spike prediction module is expressive enough to capture and replicate the underlying dynamics of the recurrent network.

## 4.2 Fully Observed Network with External Input

We next examine the quality of connectivity inference in a fully observed network subjected to external inputs, specifically synthetic, continuously varying cues designed to mimic structured angular modulation. The external drive in this setup consists of a low-amplitude signal that fluctuates within the circular space $[0, 2\pi]$, introducing gradual angular shifts in the shared input received by all neurons. The purpose of this design is to introduce structured, non-random external stimulus capable of steering the network's activity, and determine how such external stimulus influences spike predictability and connectivity inference accuracy.

To implement this, each neuron is assigned a preferred direction, arranged uniformly along a circular axis. A neuron's preferred orientation determines the direction toward which its outgoing synaptic weights are biased. The synaptic weights defined in Eq.(10) are shifted according to the external input $\theta(t)$, effectively rotating the weight matrix to align with the input direction (see Appendix G for further details). This causes the internally generated activity bumps to follow the input stimulus in synchrony, producing a smooth, input-driven trajectory across the neural manifold (Figure 3b, top row).

Empirically, we find that introducing this structured external cue leads to improved spike predictability compared to the generative network without external input, which exhibits spontaneous drift in its global activity pattern (Table 2). This suggests that networks driven by such input produce more predictable dynamics. However, despite this increase in spike predictability, connectivity inference errors remain similar or slightly elevated. This likely arise because external drive dominates spike timing, reducing the relative explanatory power of the inferred recurrent weights. Meanwhile, all baseline inference methods continue to exhibit persistent artifacts (Figure 3c, top row), such as spurious correlations between weakly or unconnected units. In contrast, our GNN-based method maintains a clear advantage, delivering more accurate reconstructions of the ground-truth network across trials (Table 2). This highlights the robustness of our approach even in settings where external inputs strongly shape network activity.

## 4.3 Weakly Correlated, Fully Observed Network with External Input

We next explore the impact of network activity correlation structure on connectivity estimation by comparing two distinct weight profiles: the local Mexican-hat profile (Figure 3a, top), which produces multiple periodic activity bumps as in Figure 3b (top), and a modified profile characterized by localized excitation at zero angular difference combined with broadly tuned inhibition (Figure 3a,

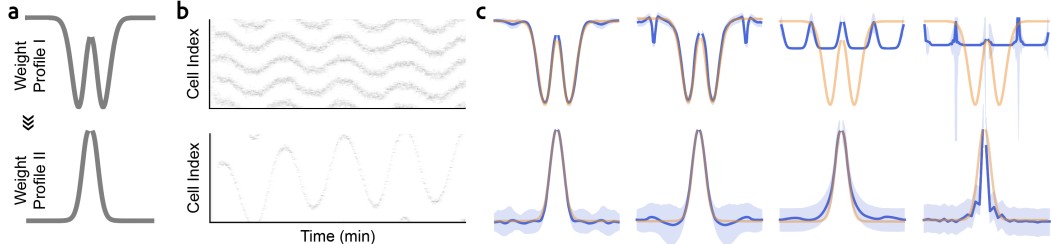

Figure 3: Evaluating connectivity inference. Top: bump dynamics in a ring attractor network driven by external input. Bottom: dynamics under a modified recurrent connectivity profile. (a) Transition from a local Mexican-hat profile (top) to a new configuration with local excitation and broadly tuned inhibition (bottom). (b) Spike raster plot showing rotating activity bumps induced by sinusoidal external inputs. (c) Comparison of ground-truth (orange) and inferred (blue) weight profiles estimated using GNN, GLM, seqNMF, and TCA, following the same order as the corresponding sub-figures.

bottom). This adjusted profile ensures the formation of only a single stable activity bump, effectively simulating conditions typical of the HD system (Figure 3b, bottom). We drive the ring network with a sinusoidal input signal that shares the same period as in Figure 3b (top), and then investigate how reducing spurious correlations in spike-train data by altering connectivity structures affects the accuracy of connectivity inference.

When shifting from the Mexican-hat to the localized excitation profile, we anticipate a notable reduction in spurious correlations among neural activities, as synchronized activity patterns diminish. This reduction is expected to lower inference errors by minimizing reliance on correlated noise and emphasizing true connectivity-driven structure. Consistent with this expectation, all baseline methods show improved inference performance under this condition (Table 3), benefiting from fewer misleading correlations. Notably, our GNN-based approach achieves even higher accuracy and robustness (Table 3 and Figure 3c, bottom [2]). These results underscore the importance of the correlation structure in shaping inference quality and reinforce the effectiveness of our method across both strongly and weakly correlated activity conditions.

Table 3: Connectivity inference error $\Delta$ and spike prediction performance $\mathcal{L}_{\text{bps}}$ for a modified input profile featuring localized excitation and broadly tuned inhibition, in a weakly correlated, fully observed network with external input. Values are averaged over three trials.

|  | Thresh. | | LNP | |
| --- | --- | --- | --- | --- |
|  | $\Delta(\downarrow)$ | $\mathcal{L}_{\text{bps}}(\uparrow)$ | $\Delta(\downarrow)$ | $\mathcal{L}_{\text{bps}}(\uparrow)$ |
| GNN | **0.048** | **2.652** | **0.043** | **2.668** |
| GLM | 0.125 | 2.534 | 0.117 | 2.576 |
| seqNMF | 0.374 | – | 0.378 | – |
| TCA | 0.362 | – | 0.369 | – |

## 4.4 Weakly Correlated, Partially Observed Network with External Input

To separate the challenges of inference in partially observed networks from those arising in strongly recurrent circuits, we have thus far focused on a fully observed setting. We now examine how varying the number and ratio of observed and hidden neurons influence spike prediction accuracy and circuit inference error within a partially observed setting, using the simulated HD network described in Section 4.3. Specifically, from a generative ring network comprising 100 neurons, we randomly select $N_o \in \{60, 80, 100\}$ observed neurons, while the remaining neurons serve as the pool for selecting $N_h \in \{0, 5, 10, 20\}$ hidden neurons. The spiking activity of the observed neurons is used as training data, and hidden neurons participate in the model as graph nodes without direct training input. By systematically varying $N_o$ and $N_h$, we assess the impact of different observed-hidden neuron configurations on circuit inference quality. When all 100 neurons are observed, hidden neurons are not included, as the total network size remains fixed at 100.

We begin by inferring synaptic connectivity among $N_o$ observed and $N_h$ hidden neurons, with each neuron assigned an angular position uniformly spaced around a ring. Although the initial inference yields connection weights between all these neurons, we retain only the weights between observed neurons, organized into a partially complete $N \times N$ weight matrix indexed by angular positions (Figure 4b). This choice reflects the transductive nature of the task: only the observed

---

[2]In Fig. 3c, the error bars in the bottom row are wider than those in the top. This can be explained as follows. Weight Profile II is a broad, smoothly varying inhibitory Gaussian. Because its tail is nearly flat, small variations in higher-frequency coefficients have little effect on predicted spikes patterns. As a result, the likelihood surface is flatter in those directions, leading to greater variability in parameter estimates across runs. Moreover, the stronger and more uniform inhibition of Profile II suppresses activity across much of the population, substantially reducing the number of informative spikes. Together, the weaker sensitivity and smaller sample of observations yield greater parameter uncertainty, which manifests as the wider error bars in the bottom row.

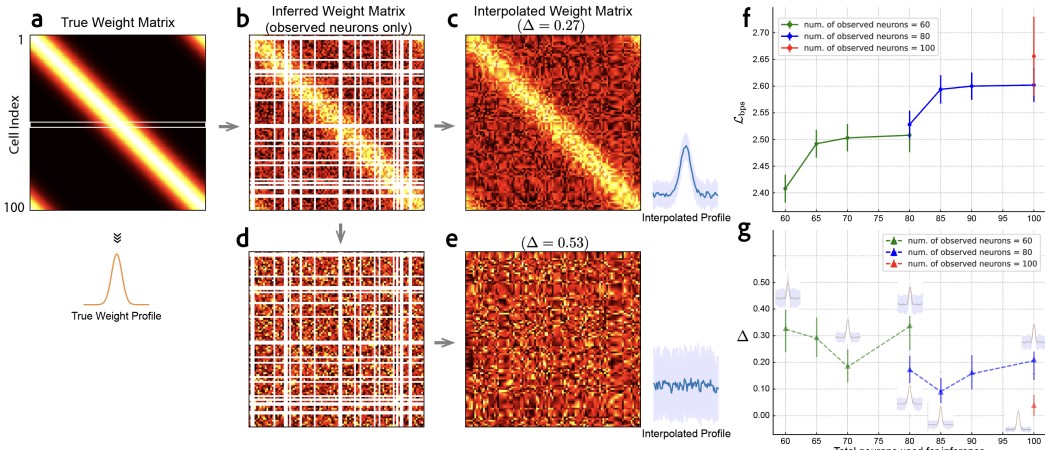

Figure 4: Evaluation of inference performance in a partially observed network as a function of the total number of neurons used for inference. (a) Ground-truth synaptic weight matrix with a smooth, spatially structured profile on a ring. (b) Synthetic weight matrix among observed neurons only ($N_o = 80$). The matrix is not inferred from spike train data but constructed by masking unobserved rows and columns of the ground-truth matrix in (a), followed by the addition of small uniform noise to mimic inference uncertainty. (c) Full weight matrix after two-dimensional interpolation, showing partial recovery of spatial structure. (d) Control: shuffled version of the inferred matrix, where either rows or columns are randomly permuted to disrupt spatial organization while preserving marginal distributions. (e) Interpolated version of the shuffled matrix exhibits degraded structure and higher error ($\Delta = 0.53$) compared to the unshuffled case ($\Delta = 0.27$). (f) Spike prediction accuracy ($\mathcal{L}_{bps}$) improves as more neurons are incorporated, with colored curves representing different numbers of observed neurons ($N_o = 60, 80, 100$). Error bars denote one standard deviation. (g) Circuit inference error ($\Delta$) plotted against the total number of neurons used for inference. Small insets show interpolated weight profiles for selected configurations, revealing how the structure quality varies with observed-to-hidden neuron ratios.

neurons have spiking activity available during training, making their inferred interactions empirically grounded. In contrast, weights involving hidden neurons rely solely on model assumptions or priors and therefore carry higher uncertainty. To reconstruct the full $N \times N$ weight matrix, including connections involving hidden neurons, we apply two-dimensional linear interpolation. This assumes that connectivity varies smoothly along the ring. To preserve the circular topology, the matrix is temporarily extended along the angular dimension. Interpolation is then applied row-wise (for outgoing connections) and column-wise (for incoming connections), filling in missing values using nearby known weights (Figure 4c).

Our analysis shows that increasing the number of observed neurons generally enhances spike prediction accuracy and concurrently reduces inference error (Figures 4f-g), indicating more precise recovery of latent dynamics when a larger fraction of the network is directly observed. When hidden neurons are added, spike prediction accuracy continues to improve across all configurations, though the gains diminish as the number of hidden neurons increases (Figures 4f). In contrast, inference error initially decreases but then saturates or slightly increases, particularly when the hidden-to-observed neuron ratio becomes large (Figure 4g). For example, with a fixed number of observed neurons, expanding the hidden population from 5 to 20 yields diminishing returns in spike prediction accuracy and can lead to a plateau or rise in inference error. This suggests that while hidden neurons can support better spike prediction by capturing latent dynamics, they may also introduce structural ambiguity, especially when weakly constrained by observed activity. These trends highlight a tradeoff between functional prediction and structural inference, and suggest that optimal performance does not necessarily result from maximizing the number of hidden neurons.

Finally, to assess whether the inferred weight matrix reflects meaningful structure beyond chance, we shuffled its rows while preserving their marginal distributions. This disrupts any spatial alignment while maintaining the local weight statistics. As shown in Figures 4d-e, the resulting interpolated matrix displays no coherent structure and yields a substantially higher inference error ($\Delta = 0.53$) compared to the unshuffled case ($\Delta = 0.27$), supporting the conclusion that the original inferred weights capture nontrivial spatial patterns not attributable to chance.

## 4.5 Head Direction Cell Network

We have demonstrated that it is possible to infer aspects of neural circuitry even when training spike data covers only a subset of the full neural population. Can this modeling framework be extended to real-world scenarios, such as analyzing neural ensembles recorded during spatial navigation or other behaviorally relevant tasks? To investigate this, we apply our circuit inference model to a publicly available dataset [36] of 19 simultaneously recorded HD cells in the anterodorsal thalamic nucleus (ADn) of freely moving mice (see Appendix E for experimental details). In this setting, we fix the total number of neurons in the ring network to $N = 100$, with $N_o = 19$ observed neurons and $N_h \in \{0, 5, 10, 20\}$ unobserved. Although the actual circuit size is unknown, choosing $N \gg N_o$ allows us to approximate the underlying connectivity with an angular resolution of $2\pi/N$.

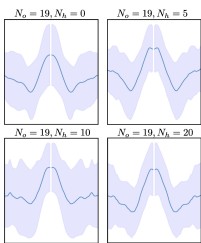

To apply the inference model, we first construct the tuning curves of the 19 observed neurons and estimate their preferred head directions (Figure 6). Each observed neuron is then matched to one of the $N$ positions in the ring network by assigning it to the neuron whose preferred head direction, spaced at intervals of $2\pi/N$, is closest to the observed tuning peak. After this assignment, $N_h$ hidden neurons are randomly selected from the remaining positions in the ring network to initialize the proposed model. Following the same training procedure as in Section 4.4, we find that the learned weight profiles consistently exhibit similar patterns across varying hidden neuron counts (Figure 5). This consistency points to a connectivity motif characterized by local excitation and surrounding inhibition, echoing the structure proposed in continuous attractor network models of HD cells [12]. Although some variability in the inferred weights is observed, likely due to the limited fraction of observed neurons relative to the full network, the results demonstrate that our framework can effectively recover key features of the underlying circuit even under partial observability in real neural recordings.

Figure 5: Inferred weight profiles derived from 19 real HD cells in the anterodorsal thalamic nucleus (ADn) of mice, with the number of observed neurons fixed at $N_o = 19$ and the number of hidden neurons ($N_h$) increasing from 0 to 20. The weights are obtained through linear interpolation under the assumption that the total number of neurons is set to 100.

## 5 Discussion

In this paper, we presented an self-supervised approach for inferring neural circuit connectivity from population spike activity. By designing two functionally specialized network modules, one for learning synaptic connectivity and the other for predicting future spiking, we established a link between the latent representations in the structure learning module and the underlying network connectivity. Using both simulated and real neural datasets, we demonstrated that our GNN-based model, which captures the dynamics of interacting neurons, accurately recovers true weight profiles and performs favorably in comparison to traditional approaches such as GLMs and other baselines that rely on approximating activity correlations. We further found that our method remains effective under diverse conditions, including varying levels of activity correlation, partial observability, and the presence or absence of external inputs. Additionally, our analysis of real neuronal data highlights the practical applicability of this approach. Specifically, applying our model to HD cell spike recordings from awake animals revealed network structures consistent with the ring attractor hypothesis proposed in earlier mechanistic studies [12].

A natural question is how the proposed framework extends beyond ring attractor networks to neural circuits with different topological structures or functional properties. The reliance on ring geometry in our current experiments arises solely from the optional geometric feature provided to the spike prediction module. The self-supervised learning objective of predicting future spikes while jointly estimating a latent connectivity matrix does not depend on this assumption. In the present implementation, the message function $\phi(\cdot)$ receives only the concatenated node embeddings and therefore imposes no explicit ring metric. When geometric priors are desirable, an additional attribute such as one-dimensional distance (chains), two-dimensional Euclidean or toroidal distance (cortical sheets, grid cells), or a learned positional embedding can be incorporated into the message function. Importantly, the choice to include or omit such features toggles the inductive bias without altering the loss, dual-module architecture, or optimization routine. This flexibility allows the same inference network to generalize across diverse circuit topologies, including those with fundamentally different structural or functional organization.

Recent approaches that couple spike modeling with latent graph estimation fall into two primary categories. NetFormer [37] encodes each neuron's recent spike history as a token and derives a step-wise attention matrix whose entries are interpreted as couplings, thereby updating the graph at every time step. Because attention weights are used directly for prediction, NetFormer lacks a dedicated message-passing module and instead recomputes pairwise interactions at each step. This design precludes iterative propagation through a learned weight matrix and limits the ability to filter indirect correlations. Notably, Fig. 15 in the original NetFormer paper reports inference performance on exactly the same benchmark considered here, and its accuracy is drastically worse than that achieved by our framework. By contrast, AMAG [38] employs message passing but initializes the adjacency matrix from random weights or correlation-based heuristics, focusing on refining rather than inferring connectivity. Our framework differs in its explicit separation of structure learning and spike prediction: a latent connectivity block continuously estimates circuit structure from activity, while a dedicated message-passing block leverages this estimate for prediction. This division yields more stable graph inference, naturally accommodates unobserved neurons through auxiliary nodes, and sustains predictive accuracy in recurrent regimes.

While these results are promising, we acknowledge a few limitations of the proposed model. First, our framework assumes that the underlying network connectivity remains fixed throughout the observation period. However, in biological neural systems, synaptic connections are often plastic and can evolve over time in response to experience or changing behavioral demands. A promising direction for future research would be to extend the model to infer time-varying connectivity, allowing for the reconstruction of dynamically changing weight profiles that better reflect the adaptive nature of real neural circuits. In addition, while our experimental setup captures the transition in the generative recurrent network from exhibiting spatially periodic activity to forming a single activity bump on the ring, this initial configuration with multiple bumps can be interpreted as a simplified representation of the grid cell (GC) system. Specifically, it resembles a scenario in which an animal moves along a direction aligned with one of the principal lattice vectors of a 2D virtual triangular lattice, thereby periodically activating every vertex along that path [39]. A natural extension of this work would be to apply our inference framework to 2D continuous attractor networks, both in simulation and using real grid cell data, to further explore its utility in inferring neural circuitry in more complex spatial systems. With the advent of multi-Neuropixels probes and mesoscale two-photon calcium imaging, it is now possible to record from hundreds to thousands of neurons across extended spatial domains. These advances open the door to deploying our approach on MEC grid cells and other spatially organized circuits with unprecedented coverage, providing a more comprehensive testbed for connectivity inference in large-scale neural populations. Finally, we acknowledge that similar neural dynamics can arise from multiple distinct circuit configurations [40, 41, 31]. Our model aims to infer one possible configuration that is consistent with the observed spiking activity, but we do not claim that the inferred circuitry is unique. Integrating tools from algebraic topology or combinatorial network theory might offer new avenues for characterizing the space of circuits compatible with observed neural activity.

Overall, our approach offers a flexible and interpretable framework for inferring latent neural connectivity from spiking data in a self-supervised and data-driven manner. By bridging structure learning with predictive modeling, we provide a general method for uncovering underlying circuit dynamics that can complement and extend existing tools in computational neuroscience.

## Acknowledgments

K.Y. is grateful to Ila Fiete and Abhranil Das for their helpful discussions during the early stages of this research, and to Xaq Pitkow for providing foundational guidance during the postdoctoral work with him. K.Y. is supported in part by the National Research Foundation of Korea (NRF) grant (No. RS-2024-00337092), the Institute of Information & communications Technology Planning & Evaluation (IITP) grants (No. RS-2020-II201373, Artificial Intelligence Graduate School Program; No. IITP-(2025)-RS-2023-00253914, Artificial Intelligence Semiconductor Support Program (Hanyang University)) funded by the Korean government (MSIT), and in part by Samsung Electronics Co., Ltd.

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

# A Generative Recurrent Network Models

We consider a ring network composed of $N$ neurons (Figure 1a), where the synaptic weights $\mathbf{W}_{ij}$ from each neuron to all others in the ring follow a rotation-invariant local Mexican-hat profile:

$$\mathbf{W}_{ij} = e^{-d_{ij}^2/2\sigma_1^2} - a e^{-d_{ij}^2/2\sigma_2^2} \tag{10}$$

This structure represents a difference of Gaussians, where $d_{ij}$ denotes the distance between neurons $i$ and $j$, and $\sigma_1$ and $\sigma_2$ are the standard deviations of the narrower and broader Gaussians, respectively, with $\sigma_1 < \sigma_2$. The parameter $a$, slightly greater than one, transforms local excitation into weak inhibition. This inhibitory effect facilitates pattern formation under a uniform feed-forward excitatory drive and maintains dynamical stability [15]. With this ring network architecture established, we explore two distinct spike generation processes.

**Threshold-crossing model** In this model, the input to each neuron at time step $t$ is determined by a weighted sum of synaptic activations from neighboring neurons, modulated by feed-forward inputs. Specifically, the input vector $\mathbf{g}(t)$ for all neurons is given by

$$\mathbf{g}(t) = r\mathbf{W}\mathbf{s}(t) + b\left(1 + \boldsymbol{\xi}(t)\right) \tag{11}$$

where $\mathbf{s}(t) \in \mathbb{R}^N$ represents the synaptic activations of $N$ neurons, and $\mathbf{W} \in \mathbb{R}^{N \times N}$ is the recurrent connectivity matrix defined in Eq.(10). The term $b$ represents a uniform excitatory drive, and $\boldsymbol{\xi}(t)$ is a white Gaussian noise term per neuron, with zero mean and a specified standard deviation, leading to Poisson-like variance proportional to the mean activation. The ratio of recurrent to feed-forward input is controlled by the recurrent weight strength parameter $r$. In this recurrent network model, a neuron $i$ emits a spike when its input $g_i(t)$ at time step $t$ exceeds a threshold $g_{th}$. The synaptic activation of spiking neurons is incremented by 1, while the activation of other neurons decays exponentially with a time constant $\tau$, following the equation:

$$\tau \frac{d\mathbf{s}(t)}{dt} + \mathbf{s}(t) = \Theta\left(\mathbf{g}(t) - g_{th}\right) \tag{12}$$

Here, $\Theta(\cdot)$ is the Heaviside step function, producing a binary vector of spikes across the network. The simulation is performed using a discrete-time update with a step size of $\Delta t$. The recurrent weight strength parameter $r$ regulates the extent of network activity correlations. When $r$ is small, feed-forward noise dominates, leading to relatively uncorrelated activity, whereas larger values of $r$ result in globally structured patterns of periodically spaced activity bumps (Figure 1b, top). The chosen value of $r = 0.025$ ensures that unconnected neurons in different co-active bumps exhibit strong correlations, making the spiking data an ideal benchmark for evaluating the performance of the proposed connectivity inference method.

**Linear-nonlinear Poisson model** The inputs to the linear-nonlinear Poisson (LNP) model are computed similarly to Eq.(11), with the key difference that the feed-forward input remains a constant $b$ across all neurons, without any additive noise. This is because stochasticity in the LNP model is inherently introduced through an inhomogeneous Poisson process:

$$\lambda_i(t) = \lambda_0 \, \mathrm{ReLU}\left[\, g_i(t) - g_{th} \,\right] \tag{13}$$

$$n_i(t) \sim \mathrm{Pois}\left(\lambda_i(t)\right) \tag{14}$$

where the firing rate $\lambda_i(t)$ of neuron $i$ is obtained by applying a rectifying nonlinearity (ReLU) to the summed input $g_i(t)$, shifted by a threshold $g_{th}$. This firing rate then governs an inhomogeneous Poisson process, which determines the number of spikes $n_i(t)$ produced by neuron $i$ at time $t$. The underlying neural dynamics remain the same as in the previous model, except that synaptic activations can now be incremented by values greater than 1, reflecting the Poisson-distributed spike count:

$$\tau \frac{d\mathbf{s}(t)}{dt} + \mathbf{s}(t) = \mathbf{n}(t) \tag{15}$$

This formulation allows for variability in spike counts at each time step, in contrast to the threshold-crossing model, where only a single spike could be emitted per neuron per time step. The exact parameter values used in both spike generation processes are listed in Table 4, and further details can be found in Das and Fiete [31].

Table 4: Model parameters of generative recurrent networks

| Parameter | Description | Value |
|---|---|---|
| $N$ | number of neurons | $10^2$ |
| $(\sigma_1, \sigma_2)$ | s.d. of two Gaussians for symmetric center-surround weights | $(6.98, 7.00)$ |
| $a$ | scalar of the second Gaussian | $1.0005$ |
| $b$ | uniform excitatory drive | $10^{-3}$ |
| $r$ | recurrent weight strength of threshold-crossing model | $2.5 \times 10^{-2}$ |
| $\sigma_\xi$ | s.d. of white Gaussian noise | $3 \times 10^{-1}$ |
| $g_{th}$ | threshold of spiking process | $7.35 \times 10^{-4}$ |
| $\tau$ | synaptic time constant | 10 ms |
| $\Delta t$ | discretization step size | 0.1 ms |

**Generative Network Parameters** The generative model used in this study corresponds to the highly structured ring attractor network operating in the strongest recurrent weight regime, as characterized in Das and Fiete [31]. The network consists of $N = 100$ neurons, with structured recurrent connectivity shaped by a symmetric center-surround profile defined by two Gaussian components. The standard deviations of the two Gaussians are $\sigma_1 = 6.98$ and $\sigma_2 = 7.00$, with a relative amplitude scaling factor $a = 1.0005$. Each neuron receives a uniform excitatory drive $b = 10^{-3}$. Recurrent interactions are governed by a coupling strength $r = 2.5 \times 10^{-2}$, consistent with the threshold-crossing model formulation. Neural activity is perturbed by white Gaussian noise with standard deviation $\sigma_\xi = 3 \times 10^{-1}$. Spiking occurs when the membrane potential exceeds the threshold $g_{\text{th}} = 7.35 \times 10^{-4}$. Synaptic dynamics are modeled with a time constant $\tau = 10$ ms, and the network is simulated using a discretization step of $\Delta t = 0.1$ ms.

# B  Extensions for External Inputs and Hidden Neurons

## B.1  Incorporating External Input

The spike prediction module, as defined by Eqs.(4)–(8), operates under the assumption of a fully observed network without external input drives. Specifically, it is designed for spike data collected from a generative network whose activity pattern undergoes noise-driven drift. However, real neural systems are often influenced by external stimuli. To account for this, our spike prediction module can be readily extended to accommodate conditions where activity patterns are driven by an external input $\theta$:

$$\mathbf{h}_{i,x}^t = f_{\text{enc},x}\left(\mathbf{x}_i^{t-\ell+1:t}\right) \quad (16) \quad \mathbf{h}_{i,\theta}^t = f_{\text{enc},\theta}\left(\theta_t - \tilde{\theta}_i - b\right) \quad (17) \quad \mathbf{h}_i^t = \left[\mathbf{h}_{i,x}^t, \mathbf{h}_{i,\theta}^t\right] \quad (18)$$

Here, $\mathbf{h}_{i,x}^t$ and $\mathbf{h}_{i,\theta}^t$ denote the initial embeddings for neuron $i$'s spike activity and the input stimulus $\theta$ at time step $t$, respectively. $f_{\text{enc},x}(\cdot)$ is essentially equivalent to $f_{\text{enc}}(\cdot)$ in Eq.(4), while $f_{\text{enc},\theta}(\cdot)$ serves as a separate encoder for $\theta$. Each neuron $i$ is assigned a base preferred stimulus value $\tilde{\theta}_i = \frac{2\pi}{N}i$, where $N$ is the total number of neurons, resulting in a uniform coverage over the circular stimulus space $[0, 2\pi)$. However, because circular variables lack an absolute origin, these preferred values are defined only up to a rotational shift. To account for this symmetry and enable alignment to any chosen reference point in the external stimulus space, we introduce a bias parameter $b$ in Eq.(17). This bias applies a global rotational shift to the population's preferred stimuli, allowing the network to align its internal representation with the external variable $\theta_t$. This design is particularly appropriate for experiments involving ring attractors that encode circular or periodic variables such as head direction (HD) or orientation[3] [42]. Ultimately, the initial state $\mathbf{h}_i^t$ of neuron $i$ is formed by concatenating both types of embeddings as shown in Eq.(18).

## B.2  Modeling with Hidden Neurons

Another critical aspect of network inference is addressing the fact that observed neural data often does not capture the entire network's activity, which is typically the case in real-world scenarios. To extend our model framework to conditions where unobserved neurons are present in the neural circuit, we introduce hidden neurons into the inference network while accounting for the absence of

---

[3]Our generative ring network in Section 3.1 produces a periodic activity pattern with multiple bumps, so it does not strictly represent the mechanistic model of the HD system, which typically features a single bump activity.

---

**Algorithm 1** Generalized Linear Model (GLM)

---

1  Initialize model parameters: $b \leftarrow 0$, $\mathbf{w} \leftarrow \mathbf{0}$, $\mathbf{z} \leftarrow \mathbf{1}$
2  **for** $i = 1$ **to** $N$ **do**
3      **for** iteration $= 1$ to $N_{\text{iter}}$ **do**
4          Compute coupling filter: $f_{\text{couple}} \leftarrow B\mathbf{z}$
5          **if** $i > 1$ **then**
6              Circularly shift $\mathbf{w}$ by $i$ steps
7          **end if**
8          Extract neighboring spike history: $\mathbf{X}_{-i} \leftarrow \mathbf{x}_{j \in N(i)}^{1:L-1}$
9          Compute log firing rate: $\log \lambda_i \leftarrow f_{\text{couple}} * (\mathbf{X}_{-i}\mathbf{w}) + b$
10         Compute loss: $\mathcal{L} = \sum_t \sum_i \left( \lambda_i^t - \mathbf{x}_i^t \log \lambda_i^t \right)$
11         Update parameters $(b, \mathbf{w}, \mathbf{z}) \leftarrow \arg\min_{b,\mathbf{w},\mathbf{z}} \mathcal{L}$ using Quasi-Newton method
12     **end for**
13 **end for**

---

their spike activity. This requires a consistent adjustment to each module. In the structure learning module, the feature embedding $\mathbf{z}_j$ of a hidden neuron $j$ is initialized by linearly interpolating the embeddings of its two nearest observed neurons on either side, using the representations derived in Eq.(1). Similarly, in the spike prediction module, the initial embedding $\mathbf{h}_j^t$ of hidden neuron $j$ is obtained via interpolation of the same two closest observed neurons' embeddings, derived from Eq.(4). Following this initialization, we update the embeddings of all neurons—both observed and hidden—through message-passing operations over the complete graph structure, while training the model exclusively on the spike activity of the observed neurons.

## C  Baseline Models

### C.1  Generalized Linear Model (GLM)

In this study, we first construct the coupling filter $f_{\text{couple}}$ as a linear combination of a set of raised cosine basis functions. Specifically, we define a basis matrix $B \in \mathbb{R}^{\frac{2\tau}{\Delta t} \times 32}$, where each column represents a distinct raised cosine filter. The coupling filter is then given by $f_{\text{couple}} = B\mathbf{z}$, with $\mathbf{z}$ denoting the basis coefficients. To compute the log firing rate $\lambda_i$ of neuron $i$, we project the spike history of its neighboring neurons onto the filter $f_{\text{couple}}$, weighted by the connection strengths $\mathbf{w}$:

$$\log \lambda_i = f_{\text{couple}} * (\mathbf{X}_{-i}\mathbf{w}) + b$$

Here, $\mathbf{X}_{-i}$ represents the spike trains from neurons in the neighborhood $N(i)$, truncated up to time $L - 1$. The symbol $*$ denotes convolution over time, capturing the temporal influence of past spikes from neighboring neurons. The resulting signal is passed through a linear transformation and offset by a bias term $b$. The parameters $b$, $\mathbf{w}$, and $\mathbf{z}$ are optimized by minimizing the negative log-likelihood through a Quasi-Newton method:

$$\mathcal{L} = \sum_t \sum_i \left( \lambda_i^t - \mathbf{x}_i^t \log \lambda_i^t \right)$$

---

**Algorithm 2** Tensor Component Analysis (TCA)

---

1 **Input:** activity tensor $\mathbf{X} \in \mathbb{R}^{N \times T \times K}$, rank $R$, #ALS steps $S = 500$
2 Smooth spikes: $\mathbf{M} \leftarrow \mathrm{EW}(\mathbf{X})$          ▷ exponentially-weighted moving average
3 Reshape $\mathbf{M}$ back to $N \times T \times K$ with $T = L/K$
4 Randomly initialize factors $\mathbf{W} = [w_n^{(r)}]$, $\mathbf{U} = [u_t^{(r)}]$, $\mathbf{V} = [v_k^{(r)}]$
5 **for** $s = 1$ **to** $S$ **do**
6     **for** $r = 1$ **to** $R$ **do**
7         Update $\mathbf{w}^{(r)} \leftarrow \arg\min_{\mathbf{w}} \| \mathbf{X}_{(1)} - \sum_r \mathbf{w} \, (\mathbf{v}^{(r)} \otimes \mathbf{u}^{(r)})^\top \|_F^2$
8         Update $\mathbf{u}^{(r)} \leftarrow \arg\min_{\mathbf{u}} \| \mathbf{X}_{(2)} - \sum_r \mathbf{u} \, (\mathbf{v}^{(r)} \otimes \mathbf{w}^{(r)})^\top \|_F^2$
9         Update $\mathbf{v}^{(r)} \leftarrow \arg\min_{\mathbf{v}} \| \mathbf{X}_{(3)} - \sum_r \mathbf{v} \, (\mathbf{u}^{(r)} \otimes \mathbf{w}^{(r)})^\top \|_F^2$
10     **end for**
11 **end for**
12 **for** $r = 1$ **to** $R$ **do**
13     $\mathbf{Z}^{(r)} = \mathbf{w}^{(r)} \otimes \mathbf{w}^{(r)} \otimes \mathbf{v}^{(r)} \otimes \mathbf{u}^{(r)}$
14     $\mathbf{E}^{(r)} = \sum_{t=1}^{T} \sum_{k=1}^{K} \mathbf{Z}_{:,:,k,t}^{(r)}$
15 **end for**
16 **Return** average interaction matrix $\mathbf{E} = \frac{1}{R} \sum_r \mathbf{E}^{(r)}$

---

## C.2   Tensor Component Analysis (TCA)

In our experiment, we recorded neural activity during a single continuous session. To apply TCA as described by Williams et al. [35], which requires data structured across multiple trials, we partitioned this continuous spike train into $K = 10$ equal-length segments. Each segment was treated as an individual trial, enabling the construction of a three-dimensional data tensor $\mathbf{X} \in \mathbb{R}^{N \times T \times K}$, where $N$ is the number of neurons, $T$ is the number of time points per segment, and $K$ is the number of segments. This tensor was then decomposed using TCA into a sum of $R$ rank-one components:

$$x_{ntk} \approx \sum_{r=1}^{R} w_n^{(r)} u_t^{(r)} v_k^{(r)}$$

In this decomposition, $w_n^{(r)}$ represents the contribution of neuron $n$ to component $r$, $u_t^{(r)}$ captures the temporal dynamics within each segment for component $r$, and $v_k^{(r)}$ accounts for variations across segments for component $r$. The factor matrices $\mathbf{W} = [w_n^{(r)}]$, $\mathbf{U} = [u_t^{(r)}]$, $\mathbf{V} = [v_k^{(r)}]$ were estimated using an alternating least squares (ALS) optimization procedure, which iteratively updates each factor while keeping the others fixed to minimize the reconstruction error. To analyze the interactions between neurons, for each component $r$, we computed the outer product:

$$\mathbf{Z}^{(r)} = \mathbf{w}^{(r)} \otimes \mathbf{w}^{(r)} \otimes \mathbf{v}^{(r)} \otimes \mathbf{u}^{(r)}$$

This four-way tensor $\mathbf{Z}^{(r)}$ encapsulates the pairwise interactions between neurons modulated by segment and temporal dynamics. Summing over the segment and time dimensions yielded a matrix $\mathbf{E}^{(r)}$ that represents the average interaction pattern for component $r$:

$$\mathbf{E}^{(r)} = \sum_{t=1}^{T} \sum_{k=1}^{K} \mathbf{Z}_{:,:,k,t}^{(r)}$$

Finally, averaging over all components provided the overall interaction matrix:

$$\mathbf{E} = \frac{1}{R} \sum_{r=1}^{R} \mathbf{E}^{(r)}$$

This matrix $\mathbf{E}$ offers a low-dimensional representation of the neural population's correlation structure, capturing both within-segment dynamics and across-segment variations.

---
**Algorithm 3** Sequence Non-negative Matrix Factorization (seqNMF)
---

1 **Input:** Data matrix $\mathbf{X} \in \mathbb{R}^{N \times T}$; number of components $K$; sequence length $L$; regularization parameter $\lambda$; smoothing matrix $\mathbf{S} \in \mathbb{R}^{T \times T}$ with $\mathbf{S}_{i,j} = 1$ if $|i - j| < L$, else 0

2 **Initialize:** $\mathbf{W} \in \mathbb{R}^{N \times K \times L}$ and $\mathbf{H} \in \mathbb{R}^{K \times T}$ with non-negative values

3 **for** iteration $= 1$ to $N_{\text{iter}}$ **do**

4       Compute reconstruction: $\tilde{\mathbf{X}} = \sum_{k=1}^{K} \mathbf{W}_{:,k,:} * \mathbf{H}_{k,:}$

5       Compute cross-correlation matrix: $\mathbf{C}_{k,k'} = \sum_{l=1}^{L} \sum_{n=1}^{N} \mathbf{W}_{n,k,l} \cdot \mathbf{X}_{n,t+l}$ for all $k \neq k'$

6       Update $\mathbf{W}$ and $\mathbf{H}$ by minimizing:

$$\mathcal{L} = \left\| \mathbf{X} - \tilde{\mathbf{X}} \right\|_F^2 + \lambda \sum_{k \neq k'} \left\| \mathbf{C}_{k,k'} \cdot \mathbf{S} \cdot \mathbf{H}_{k',:}^{\top} \right\|_1$$

7       using multiplicative gradient descent

8 **end for**

9 **for** $k = 1$ to $K$ **do**

10       Compute component tensor: $\mathbf{Z}^{(k)} = \mathbf{W}_{:,k,0} \otimes \mathbf{W}_{:,k,1} \otimes \mathbf{H}_{k,:}$

11       Compute interaction matrix: $\mathbf{E}^{(k)} = \sum_{t=1}^{T} \mathbf{Z}^{(k)}_{:,:,t}$

12 **end for**

13 Compute average interaction matrix: $\mathbf{E} = \frac{1}{K} \sum_{k=1}^{K} \mathbf{E}^{(k)}$

14 **Output:** Factors $\mathbf{W}$, $\mathbf{H}$, and interaction matrix $\mathbf{E}$

---

## C.3 Sequence Non-negative Matrix Factorization (seqNMF)

To extract repeated temporal motifs from high-dimensional neural recordings, we apply seqNMF, a regularized form of convolutional NMF introduced by Mackevicius et al. [34]. Given a neural activity matrix $\mathbf{X} \in \mathbb{R}^{N \times T}$, where $N$ is the number of neurons and $T$ is the number of time points, seqNMF decomposes $\mathbf{X}$ into $K$ components via a convolutional model:

$$\tilde{\mathbf{X}} = \sum_{k=1}^{K} \mathbf{W}_{:,k,:} * \mathbf{H}_{k,:}$$

Here, $\mathbf{W} \in \mathbb{R}^{N \times K \times L}$ encodes the sequence templates of length $L$, and $\mathbf{H} \in \mathbb{R}^{K \times T}$ specifies when each template is active. The convolution $*$ operates over the temporal axis. The model is fit by minimizing the objective:

$$\mathcal{L} = \left\| \mathbf{X} - \tilde{\mathbf{X}} \right\|_F^2 + \lambda \left\| \mathbf{W}^{\top} \mathbf{X} \cdot \mathbf{S} \cdot \mathbf{H}^{\top} \right\|_{1, i \neq j}$$

where $\mathbf{S} \in \mathbb{R}^{T \times T}$ is a smoothing matrix with ones on a band of width $L = 2$ around the diagonal, and $\lambda = 0.001$ controls the strength of a regularization term that penalizes redundancy across components. The factors $\mathbf{W}$ and $\mathbf{H}$ are updated via multiplicative gradient descent [43]. After training, the interaction structure between neurons is estimated from each sequence component. For each $k$, we compute a correlation tensor:

$$\mathbf{Z}^{(k)} = \mathbf{W}_{:,k,0} \otimes \mathbf{W}_{:,k,1} \otimes \mathbf{H}_{k,:}$$

Then, summing over time yields:

$$\mathbf{E}^{(k)} = \sum_{t=1}^{T} \mathbf{Z}^{(k)}_{:,:,t}, \quad \text{and} \quad \mathbf{E} = \frac{1}{K} \sum_{k=1}^{K} \mathbf{E}^{(k)}$$

This provides a low-dimensional representation of temporally organized pairwise neural interactions.

## D   Comparison with the GLM

The GLM [1] is a widely used approach for modeling neural spike trains. While it has proven effective for predicting neural activity and estimating coupling filters, its formulation differs fundamentally from our proposed framework in several respects.

In the GLM, the weight matrix $\mathbf{w}$, which governs pairwise influences, is directly parameterized and optimized from a random initialization via log-likelihood maximization. There is no separate function or intermediate representation devoted to inferring structure. In contrast, our model introduces a dedicated structure learning module that explicitly extracts latent embeddings of neuron spike trains and predicts edge strengths using an MLP. This step isolates the task of estimating connectivity and makes it learnable independently of the spike generation process, enabling the model to reason about connectivity patterns beyond what improves immediate likelihood fit.

Moreover, the GLM computes the log firing rate of each neuron as a linear projection of the temporally filtered spike history of its neighbors onto the coupling filters. This process lacks a latent embedding space or dynamic aggregation step. Crucially, the influence from other neurons is not passed via abstract latent messages, but directly via convolved spike histories. This approach enforces an additive, filter-based influence model with limited capacity to capture nonlinear or recurrent interactions. In contrast, our spike prediction module performs message passing on top of learned embeddings. Each neuron's spike history is first embedded into a latent state, messages are computed between neuron pairs based on these embeddings, and then dynamically integrated using a gated recurrent unit. This framework supports richer, nonlinear temporal dependencies and is known to be capable of modeling a wide class of dynamical systems.

In summary, the key distinction lies in the explicit modularity of our model and its reliance on latent, dynamically integrated representations, which enable it to capture and refine structural hypotheses based on rich temporal dependencies. This sets it apart from the GLM, which lacks both the architectural separation and the representational flexibility to perform such inference.

## E   Head Direction Cell Benchmark

The head direction cell benchmark dataset, available from CRCNS (Collaborative Research in Computational Neuroscience), contains extracellular recordings from the antero-dorsal thalamic nucleus and post-subiculum of freely moving mice as they foraged for scattered food in a $53 \times 46$cm open arena, specifically designed for analyzing head direction (HD) cell dynamics [36]. Although not every recorded neuron exhibits HD properties, a session was chosen based on having the highest number of neurons meeting a HD score threshold, resulting in 19 simultaneously recorded HD cells (Figure 6). The HD scores were computed by binning head direction data into $3°$ intervals, computing firing rates within these bins, smoothing the results with a $14.5°$ boxcar filter, and calculating the Rayleigh vector length from the resultant tuning curves. The threshold for identifying HD cells was defined using the 99th percentile of a null distribution generated by by shuffling spike trains. To avoid over-representation of any particular angle on the ring network, sessions were further screened to minimize the KL divergence between each session's preferred direction distribution and a uniform distribution. These steps produce a dataset well-suited for inferring underlying circuits of HD neurons.

## F   Evaluation Metrics

To evaluate a weight matrix where each neuron's weight vector (i.e., each row) is expected to exhibit rotational invariance, the first step is to align all rows to a common reference phase. This is necessary because the weight vectors can be interpreted as the same underlying pattern, each shifted by a different rotation. Therefore, each row must be shifted so that all vectors share a consistent phase. Additionally, the inferred weights may differ from the true weights by an arbitrary global scale factor—this can result from the multiplicative interaction between connection strength and incoming messages in Eq.(6). Since such scaling should not affect the evaluation, we first compute the average aligned weight vector $\bar{\mathbf{w}}$ across neurons. We then learn a scale factor that minimizes the $\ell_1$-distance between $\bar{\mathbf{w}}$ and the true weight profile (e.g., a representative row from $\mathbf{W}$), and apply this same scaling to the entire inferred matrix $\mathbf{w}$ to obtain $\hat{\mathbf{W}}$. Finally, we assess the quality of the inferred weights using the normalized inference error [31], given by $\Delta = \frac{\|\mathbf{W}-\hat{\mathbf{W}}\|_F}{\|\mathbf{W}\|_F}$, where $\|\cdot\|_F$ denotes the Frobenius norm.

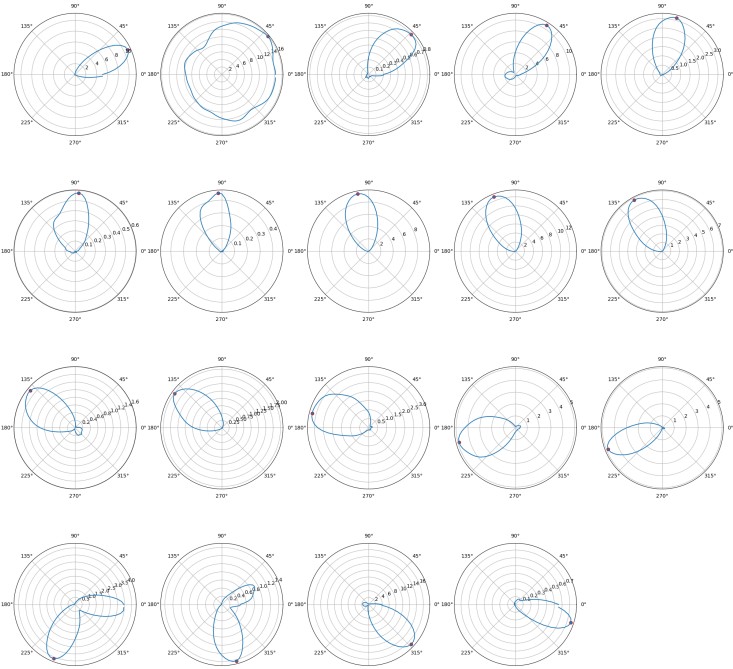

Figure 6: Tuning curves of 19 simultaneously recorded HD cells that passed the HD cell threshold criteria. Each polar plot represents the directional tuning of a single HD cell, showing firing rate as a function of head direction. The cells displayed here include those with the most widely distributed preferred directions across all sessions and animals, highlighting the diversity of head direction representations within the recorded population. Red dots indicate the peak firing direction (preferred head direction) for each cell.

As a separate measure from connectivity inference, we evaluate spike prediction performance using a log-likelihood-based metric that compares our model to a homogeneous Poisson process [44, 45, 1]. This metric quantifies how much better the model predicts spike timing relative to a baseline that assumes a constant firing rate. A higher score indicates that the model more accurately captures the temporal structure of neural activity. Specifically, the metric is defined as

$$
\mathcal{L}_{\text{bps}} = \frac{1}{N} \sum_{i=1}^{N} \frac{\sum_{t=1}^{T} \left[ (x_i^t \log \lambda_i^t - \lambda_i^t) - \left( x_i^t \log \bar{\lambda}_i - \bar{\lambda}_i \right) \right]}{\sum_{t=1}^{T} x_i^t}
\tag{19}
$$

where $\bar{\lambda}_i = \frac{1}{T} \sum_{t=1}^{T} x_i^t$ represents the empirical average firing rate of neuron $i$. By normalizing the log-likelihood improvement by the total number of spikes, this metric yields an interpretable value in bits per spike. The final score $\mathcal{L}_{\text{bps}}$ is obtained by averaging across all neurons.

## G  Externally Driven Rotation of Activity Bumps in a Ring Network

To incorporate external cues—such as head direction, denoted by $\theta(t)$—into a ring attractor model, we adapt the synaptic connectivity to dynamically reflect changes in the input. In the standard ring attractor, stationary activity bumps are maintained through symmetric recurrent connections. Here, we introduce a mechanism in which the synaptic weight matrix is shifted in real time based on the external signal $\theta(t)$, causing the activity bumps to rotate along the ring in synchrony with the stimulus.

Each neuron $i \in \{0, 1, \ldots, N-1\}$ is assigned a preferred angle $\phi_i = \frac{2\pi i}{N}$, forming a uniform circular arrangement. This setup allows us to interpret neuron indices as discrete angular positions, facilitating a direct mapping between the continuous angular domain and its discrete neural implementation. The synaptic weight from neuron $i$ to neuron $j$, adjusted by the input angle $\theta(t)$, is given by:

$$
W_{ij}^{(\theta)} = W_0 \left( d(\phi_i - \phi_j - \theta) \right), \quad d(\alpha) = \text{mod}(\alpha + \pi, 2\pi) - \pi
$$

where $d(\cdot)$ computes the signed shortest angular distance. The base connectivity profile $W_0(\delta)$ is modeled using a Mexican-hat kernel:

$$W_0(\delta) = \exp\left(-\frac{\delta^2}{2\sigma_1^2}\right) - a\exp\left(-\frac{\delta^2}{2\sigma_2^2}\right)$$

In practice, the rotation induced by $\theta(t)$ is implemented by translating neuron indices via integer shifts: the continuous input is scaled to a discrete offset $\Delta(\theta) = \left\lfloor \frac{N \cdot g \cdot \theta(t)}{2\pi} \right\rfloor$, where $g$ is a gain factor that controls how strongly the external input steers the bump position. The weights are then indexed as $(j - \Delta(\theta)) \bmod N$ to maintain circular continuity. This shifting scheme results in a coherent translation of the activity bumps that tracks the external input $\theta(t)$ while preserving the internal activity profile (see Figure 3b). The network thus transitions from a self-sustained attractor to one that is steerable, integrating structured external input with internally generated dynamics.

## H    Details of Compute Resources

All experiments in this study required approximately 7 GPU days on NVIDIA V100 32GB GPUs. Both the generative recurrent networks and the inference models were implemented using the PyTorch [46] and PyG [47] libraries. While the inference model is relatively lightweight to train, generating synthetic spike trains from recurrent dynamical models can take several days depending on network size and simulation length. Nonetheless, overall compute demands remain modest and accessible.

