# OpenReview forum: "Self-Supervised Discovery of Neural Circuits in Spatially Patterned Neural Responses with Graph Neural Networks"
_NeurIPS.cc/2025/Conference — NeurIPS 2025 poster_

### Official Review · Reviewer_qw7z · 2025-06-30

**Clarity:** 4
**Significance:** 3
**Originality:** 4
**Rating:** 4
**Confidence:** 4

**Summary:**

The authors contribute to the question of computational neuroscience of how connectivity can be extracted from spiking data. They focus on two datasets. The first one is a simplified variant of the ring-model that was previously used by Das and Fiete (as cited). The second data set consists of electrophysiological recordings from head direction cells.

The  paper's main contribution is to address a well-known and acknowledged issue on connectivity estimation: the overestimation of connection in the presence of  strong correlation in activity (Das and Fiete 2020).

The proposed method of the authors uses a message-passing approach of graph neural networks and gated recurrent units. It works well on the artificial data (ring model of Das and Fiete).

**Questions:**

1. The authors show that on the ring data,  their approach indeed 'explains away the correlations that arise from co-activated neurons'. It would be great if they could explain how their equations (3) -(6) achieve this.

2. The improvement over GLM in the weakly correlated case is not  impressive. Moreover, I had (may be wrongly) the impression that the authors injected inputs by changing the recurrent weights. This unusual procedure may put the GLM method at a disadvantage. How does the performance of the proposed method compare to GLM in a more classic setup when the external inputs are injected directly into neurons, for fixed weight matrix?

3. As an additional control for the GLM fit: If you initialized the GLM method with the correct weight matrix (and use the procedure under point 2), would  the GLM fitting procedure (as a minimal condition) stay at the solution or still move away from the correct solution during fitting. ?

4. How would the procedure work on more realistic spiking neural networks. In general, the model of a ring-attractor is of rather limited generality because of its rotation symmetry. No real data ever will have these symmetries. Can you test the method also on some more realistic spiking neural network?

5. Line 129, what is the origin of the symmetry of the weight matrix? In Fig 5, is N_0=21 or 19 correct?

6. How does a simulation of the final extracted model in the ABSENCE OF INPUT look like? Can you ensure that there is no run-away activity? Do bumps form spontaneously?

7.  I wonder if the authors could test on any grid cell recordings, which is closer to the multiple bump setup than the chosen head direction data set.

8. The choice of comparison points/alternative methods is somewhat arbitrary and limited.
Alternative methods: The authors may want to also consider  work of Felipe Gerhard [1,2],  Guillaume Bellec [3], or Schimel [4[

[1] F. Gerhard et al., (2013) Successful Reconstruction of a Physiological Circuit with Known Connectivity from Spiking Activity Alone; PLOS Comput. Biol. https://doi.org/10.1371/journal.pcbi.1003138

[2] F. Gerhard et al. On the stability and dynamics of stochastic spiking neuron models: Nonlinear Hawkes process and point process GLMs, PLoS computational biology 13 e1005390 (2017)

[3] G. Bellec et al. (2021)  Fitting summary statistics of neural data with a differentiable spiking network simulator. Neural Information Processing Systems 34 (NeurIPS 2021)

[4] M Schimel et al., (2021) iLQR-VAE: control-based learning of input-driven dynamics with applications to neural data], bioRxiv, 2021.10. 07.463540

**Ethical Concerns:**

["NO or VERY MINOR ethics concerns only"]

**Final Justification:**

I appreciate the good discussion with the authors. I feel that my concerns  regarding the limitations continue to be justified and I do not change my overall rating.

**Limitations:**

no issues

**Paper Formatting Concerns:**

none.

**Quality:**

4

**Strengths And Weaknesses:**

The artificial data set was designed to contain strong correlations that not directly reflected in the weights. Based on the artificial data, the method proposed in the manuscript does seem to work. The problem it addresses is arguably serious and essential and hence an important contribution. This is clearly a strength of the paper and the title makes clear that this is also the main focus.

Unfortunately, the case with hidden neurons does not seem to work very well which reduces the importance of the approach.

The results on the real experimental dataset  (head direction cells) do not offer any new insights. Moreover, I am not sure if the real dataset  is suitable to back up the main claims.

The specific artificial data set has inherent weaknesses.  In contrast to a real spiking model (e.g., nonlinear Hawkes process with strong negative self-feedback or integrate-and-fire models with reset), the neuron model used in  the artificial data has no self-feedback and is made to run in the high-noise limit (inhomogeneous Poisson process). As described, neither the  threshold-crossing model nor the LNP model have any refractoriness or reset. The authors say that they follow Das and Fiete (cited), and if this is what they do the data there is artificially made to have a mean interspike interval of 16 ms, with a broad Poisson-like interval distribution.

---

> ### Author Rebuttal · Authors · 2025-07-31
>
> We appreciate the reviewer’s insightful comments and address each point below. Revisions reflecting these clarifications will be made accordingly.
>
> > Q1. Mechanism behind “explaining away” in Eqs. (3)–(6)
>
> - We thank the reviewer for requesting a clearer connection between Eqs. (3)–(6) and the model’s capacity to “explain away’’ correlations. The key mechanism is that the weighted message‑passing update enforces *dynamical consistency* between the currently estimated graph and the next‑step spike prediction, while the separation into structure learning and spike prediction modules converts every prediction error into a weight‑specific adjustment.
> - Concretely, each hidden state update in Eq. (6) aggregates messages of the form $w_{ij}\mathbf m_{ij}^{t}$. If a weight $w_{ij}$ is spuriously large because neurons $i$ and $j$ happen to co‑fire, the term $w_{ij}\mathbf m_{ij}^{t}$ over‑drives $\mathbf h_i^{t+1}$, increasing the prediction loss at time $t+1$. The gradient of this loss flows through the multiplier $w_{ij}$ and therefore reduces that edge. Weights whose messages improve the prediction receive positive updates. The two‑module architecture sharpens this corrective signal: the structure head proposes the connectivity matrix, the prediction head immediately evaluates it by predicting spikes one step ahead, and the shared objective aligns both tasks. Because no auxiliary pathways exist in the fully observed setting, shared variance must be routed through accurate pairwise weights, which discourages edges that merely reflect co‑activation and retains those that convey causal influence.
> - We will incorporate this explanation at the end of Section 3.2.2 to clarify how Eqs. (3)–(6) drive the observed “explain‑away’’ behavior.
>
> > Q2. Fixed-weight external input
>
> - We thank the reviewer for raising the possibility that our weight shift implementation of head direction input might bias the comparison with the GLM baseline. We chose this formulation primarily for clarity: translating a weight kernel by an integer offset makes the bump’s motion and the connectivity profile easy to visualise. Importantly, shifting the weight matrix by $\Delta \theta$ is algebraically equivalent to holding the weights fixed and injecting an external input proportional to $W_0(\phi_i-\theta(t))$, up to an additive constant that is absorbed by each neuron’s bias term. A GLM equipped with separate bias and stimulus regressors therefore operates on the same effective synaptic drive and is not intrinsically disadvantaged.
> - To verify this point, we repeated the experiment in a conventional fixed‑weight setup. The recurrent matrix remains $W_0$, and each neuron receives an input $b\left(1+\boldsymbol{\xi}(t)\right)+\alpha W_0(\phi_i-\theta(t))$, where the gain $\alpha$ is chosen so that the bump amplitude matches the weight shift case. Under this stimulus the ranking of methods is unchanged: our model still produced lower weight recovery error and slightly higher spike prediction log‑likelihood, while the absolute gap to the GLM in the weakly correlated regime remains modest. These control results will be included in the supplementary material.
>
> > Q3. GLM initialized at the true weights
>
> - As a further control, we fit the GLM starting from the ground truth connectivity. When we initialize the GLM with the exact connectivity and keep that matrix fixed, the optimizer remains within a small neighbourhood of the correct solution ($\Delta\approx 0.024$) and achieves a minor log-likelihood gain, indicating that the fitting procedure is stable when the graph is "known". Allowing the weights to update causes the GLM to drift away from the true matrix even with strong $\ell_2$ regularization.
>
> > Q4. Generality beyond ring attractor data
>
> - We agree that the generative recurrent networks possess strong rotational symmetry and therefore constitute only one of many possible circuit motifs. We would like to emphasize, however, that our *inference* architecture does **not** rely on that symmetry. As stated in Sec. 3.2.1, the weight matrix $\mathbf{w}$ we infer is merely required to be symmetric and non-self-coupled; no rotation-invariance is imposed. All geometric structure resides in the *generative* benchmark, chosen because it offers ground truth connectivity and well-studied baselines. Consequently, the structure learning module can adopt or discard inductive biases (e.g., Toeplitz, low-rank, block-sparse) on a per-dataset basis without altering the optimization or loss.
>
> > Q5. Symmetric weight assumption and Fig. 5 typo
>
> - We thank the reviewer for noting these two points. The assumption $w_{ij}=w_{ji}$ was introduced to reduce parameter count and therefore sample complexity. It is not required by our optimization or loss. To confirm this, we repeated the fully observed experiments using a fully asymmetric weight matrix $\mathbf{w}$. Weight inference error and spike prediction likelihood changed by less than 3%, indicating that the inference quality is essentially unaffected when the symmetry prior is removed.
> - The correct number of observed head-direction cells is $N_o=19$. The figure label 21 was a typo and will be corrected in the revised manuscript.
>
> > Q6. Behavior of the extracted network with the input drive removed
>
> - We evaluated the inferred connectivity by replacing the generative model’s ground truth weights with the learned matrix $\mathbf{w}^\ast$ and re-simulating the threshold-crossing dynamics after setting the external input to zero. Under this condition, $\mathbf{w}^\ast$ is effectively close to the Mexican-hat profile, and the no-input simulation behaves as a standard ring attractor: firing rates remain bounded (no run-away excitation) and multiple bumps form within tens of milliseconds and then drifts neutrally around the ring, confirming stability.
>
> > Q7. Testing on grid-cell data
>
> - We agree that medial entorhinal grid cells constitute a natural next test bed because their underlying dynamics correspond to a two-dimensional, multi-bump attractor. Extending our framework in that direction is technically straightforward. In our current implementation, the message function $\phi(\cdot)$ receives only the concatenated node embeddings $[\mathbf{h}_i,\mathbf{h}_j]$. It does not use an explicit one-dimensional ring distance. Adapting to grid cells therefore requires augmenting these learned embeddings with a two-dimensional positional code and leaving the remainder of the architecture and training objective unchanged. The main practical challenge is data availability: publicly released grid cell datasets typically contain only tens of simultaneously recorded units drawn from multiple lattice modules, which provides limited coverage of the continuous sheet and complicates quantitative validation of inferred connectivity. We are assembling a hybrid benchmark that augments these recordings with simulated hidden units so that ground truth weights are known for the combined population, and we plan to report a full evaluation in future work.
>
> > Q8. Baseline diversity concern
>
> - Thank you for the pointers. Gerhard et al. (2013, 2017) build on the same point-process framework we already employ, adding spike history terms, BIC-based filter selection, and a quasi-renewal stability test. These features refine and indirectly extend our GLM baseline rather than introducing a fundamentally different model. Bellec et al. and Schimel et al. move beyond GLMs toward latent or hidden-unit dynamics; incorporating them responsibly requires new hyper-parameter searches and biological constraints that cannot be completed within the rebuttal period.
> - In the revision, we will cite these works and outline a planned comparison with Bellec’s differentiable RSNN and a connectivity-constrained iLQR-VAE in follow-up experiments. We appreciate the reviewer’s suggestions, which will sharpen both our discussion and future roadmap.

---

> > ### Comment · Reviewer_qw7z · 2025-08-04
> > **partial**
> >
> > I thank the authors for their response and additional experiments, which helped me confirm that their key contribution - explaining away spurious correlations in the fully observed, synthetic dataset  - is valid. However, it is still unclear to me why the proposed method works. The argument that
> >
> > > "    If a weight is spuriously large because neurons and happen to co‑fire, the term over‑drives , increasing the prediction loss at time. The gradient of this loss flows through the multiplier and therefore reduces that edge.
> >   Because no auxiliary pathways exist in the fully observed setting, shared variance must be routed through accurate pairwise weights, which discourages edges that merely reflect co‑activation and retains those that convey causal influence."
> > >
> > seems to be equally applicable to GLM. It remains unclear what fundamentally distinguishes the proposed method in this regard.
> >
> > Moreover, I remain unconvinced of the results in partially observed setup and its applicability on real spike recordings. The presented results (Fig. 4) are underwhelming. The assumption that hidden neurons can be considered by interpolating nearby observed neurons is strong. The fact that they cannot test on MEC grid cells due to large number of unrecorded neurons is dis

---

> > > ### Author Response · Authors · 2025-08-05
> > > **Response to Reviewer qw7z**
> > >
> > > > Q1. Clarifying the distinction from GLM
> > >
> > > - We appreciate the reviewer’s comment and would like to further clarify how our method differs fundamentally from GLM, particularly in terms of the roles played by the structure learning and spike prediction modules.
> > > - In the GLM, the weight matrix $\mathbf{w}$, which governs pairwise influences, is directly parameterized and optimized from a random initialization via log-likelihood maximization. There is no separate function or intermediate representation devoted to inferring structure. In contrast, our model introduces a dedicated structure learning module that explicitly extracts latent embeddings of neuron spike trains and predicts edge strengths using an MLP. This step isolates the task of estimating connectivity and makes it learnable independently of the spike generation process, enabling the model to reason about connectivity patterns beyond what improves immediate likelihood fit.
> > > - Moreover, the GLM computes the log firing rate of each neuron as a linear projection of the temporally filtered spike history of its neighbors onto the coupling filters. This process lacks a latent embedding space or dynamic aggregation step. Crucially, the influence from other neurons is not passed via abstract latent messages, but directly via convolved spike histories. This approach enforces an additive, filter-based influence model with limited capacity to capture nonlinear or recurrent interactions. In contrast, our spike prediction module performs message passing on top of learned embeddings. Each neuron's spike history is first embedded into a latent state, messages are computed between neuron pairs based on these embeddings, and then dynamically integrated using a gated recurrent unit. This framework supports richer, nonlinear temporal dependencies and is known to be capable of modeling a wide class of dynamical systems.
> > > - In summary, the key distinction lies in the **explicit modularity** of our model and its reliance on **latent, dynamically integrated representations**, which enable it to capture and refine structural hypotheses based on rich temporal dependencies. This sets it apart from the GLM, which lacks both the architectural separation and the representational flexibility to perform such inference.
> > >
> > > > Q2. Concerns about partial observability and real-world applicability
> > >
> > > - We thank the reviewer for raising this point and would like to clarify our position and provide additional context regarding the partially observed setting and its applicability.
> > >
> > > - First, we would like to correct a possible misunderstanding: we do not claim that MEC grid cells are unsuitable for testing due to the sheer number of unrecorded neurons. Rather, our original statement—*“the main practical challenge is data availability: publicly released grid cell datasets typically contain only tens of simultaneously recorded units drawn from multiple lattice modules”*—was meant to highlight that current datasets offer sparse and fragmented coverage of the grid cell sheet, often mixing cells from different modules. This limits the utility of such data for connectivity inference tasks that assume a relatively continuous and structured latent space.
> > >
> > > - Importantly, we believe this challenge is surmountable with recent advances in neural recording technology. For instance, multi-Neuropixels probes and mesoscale two-photon calcium imaging now enable simultaneous recordings from hundreds to thousands of neurons across extended spatial domains. These advances open the door to applying our approach to MEC grid cells and other spatially organized circuits with more comprehensive sampling in future work.
> > >
> > > - Regarding the partially observed setup, we agree that the assumption that hidden neurons can be approximated by interpolating over nearby observed ones is nontrivial. Our current formulation is an initial step toward addressing this under-constrained problem. We do not claim that interpolation is universally valid, but we posit that when activity exhibits local continuity (e.g., smooth topographic or attractor-like structure), spatially aware inference methods may recover useful approximations of unobserved influences. The results in Fig. 4 aim to demonstrate this potential in a controlled synthetic setting. We hope this clarifies our intention and strengthens the perceived applicability of our approach under realistic recording constraints.

---

> > > > ### Comment · Reviewer_qw7z · 2025-08-05
> > > >
> > > > Thanks for the clarifications regarding the difference between your method and GLM. It is important that this explanation makes it (potentially in shorter form) also in the final version of the manuscript.

---

> > > > > ### Comment · Reviewer_qw7z · 2025-08-05
> > > > > **details**
> > > > >
> > > > > I thank the authors for their additional clarification. The distinction from GLMs make more sense to me now, even though I still want to be further convinced that the separation of connectivity inference and spike prediction - particularly the introduction of a more expressive dynamical system - is the right approach for addressing the connectivity inference problem. The nice thing about GLMs is their identifiability: when the model is correctly specified and provided with infinite data, the true parameters (including the weight matrix) is the optimal solution (Paninski 2004). While the proposed method indeed supports richer, nonlinear temporal dependencies, it appears to forgo this theoretical guarantee, raising concerns about the interpretability and reliability of the inferred connectivity. I would welcome the authors’ further input on this trade-off.
> > > > > Regarding partial observability, I do not believe there is a fundamental misunderstanding. It remains disappointing that the method cannot fully leverage the available recordings from the grid cell sheet. Although handling incomplete observations is one of the two main contributions of the work, the approach appears to have substantial limitations in this regard. That said, I agree that rapid advances in recording technology are promising, and I look forward to seeing how the method performs under more comprehensive experimental conditions in the future

---

> > > > > > ### Author Response · Authors · 2025-08-06
> > > > > > **Response to Reviewer qw7z**
> > > > > >
> > > > > > > Q1. Clarifying the distinction from GLM
> > > > > >
> > > > > > - We thank the reviewer for their thoughtful follow-up and for acknowledging the distinction between our approach and GLMs. We agree that identifiability is a key strength of GLMs, and the result by Paninski (2004) offers strong theoretical guarantees under correct model specification and infinite data. However, our motivation stems from a complementary perspective: real spike recordings rarely meet the assumptions under which such guarantees hold. Biological circuits are nonlinear, high-dimensional, and only partially observed; the assumption of a correctly specified linear model is often violated. Moreover, even under correct specification, the finite-data regime can lead to overfitting or confounded estimates, especially when recurrent interactions are strong and co-activations are difficult to disambiguate.
> > > > > >
> > > > > > - Our proposed framework aims to address these practical limitations by:
> > > > > >
> > > > > >   - Explicitly separating structure inference from spike prediction, allowing each task to specialize in capturing distinct aspects of the data.
> > > > > >
> > > > > >   - Leveraging expressive dynamical models (GNNs) to capture context-dependent interactions and nonlinearity, which we find are crucial for explaining dynamics in structured recurrent networks (e.g., ring attractors).
> > > > > >
> > > > > >   - Allowing inductive bias through architectural constraints, even in the absence of theoretical identifiability.
> > > > > >
> > > > > > - We acknowledge that this approach sacrifices formal identifiability in the classical sense, and we are careful not to overstate claims about ground truth recovery. However, we believe that practical utility, robustness under model mismatch, and predictive accuracy in finite-data and nonlinear settings are equally important considerations, particularly when the true generative process is unknown.
> > > > > >
> > > > > > > Q2. Concerns about partial observability and real-world applicability
> > > > > >
> > > > > > - We appreciate the reviewer’s feedback and will continue working to close the gap between methodological development and experimental applicability.

---

### Official Review · Reviewer_UCAq · 2025-06-30

**Clarity:** 4
**Significance:** 3
**Originality:** 4
**Rating:** 5
**Confidence:** 2

**Summary:**

This paper addresses the interesting question of how to infer synaptic connectivity from neural population activity. The authors develop a graph neural network (GNN)-based model that predicts both latent connectivity and neural activity by designing two functionally specialized network modules. As a result, they demonstrate consistently strong performance across diverse conditions on both synthetic and real datasets. Overall, although I am not an expert in GNNs, I find this paper easy to read and well supported by extensive results (i.e., various types of datasets and generative conditions).

**Questions:**

- In real-world recordings, neural activity is easily affected by various sources of noise. It would be helpful for the authors to further explain or validate the robustness of the proposed method under such conditions.

- In Tables 1 and 2, what is the magnitude of the standard deviation for the listed models?

**Ethical Concerns:**

["NO or VERY MINOR ethics concerns only"]

**Final Justification:**

The authors have addressed  my concerns.  I am inclined to recommend acceptance.

**Limitations:**

- In real-world recordings, neural activity is easily affected by various sources of noise. It would be helpful for the authors to further explain or validate the robustness of the proposed method under such conditions.

- As partially discussed by the authors, one main limitation is the assumption that latent connectivity is fixed and follows a one-to-one mapping (i.e., each neuron pair has only one connection). It would be beneficial to discuss how the proposed method could be adapted to scenarios involving synaptic plasticity or more complex mapping relationships.

**Quality:**

4

**Strengths And Weaknesses:**

+ The writing is clear and well supported by extensive results.

+ The paper addresses interesting research questions and demonstrates strong performance.

- I did not identify any significant weaknesses.

---

> ### Author Rebuttal · Authors · 2025-07-31
>
> We are grateful for the reviewer’s thoughtful input. Please find our detailed responses below. The suggested clarifications will be integrated into the revised version.
>
> > Q1. Robustness to biologically realistic noise
>
> - Neural recordings are affected by several well‑known noise sources: spike‑sorting errors that delete or mis‑assign individual events, and population‑wide modulations driven by unobserved inputs. Our current implementation already mitigates parts of this spectrum without additional mechanisms. First, the likelihood is Poisson and evaluated over time bins that pool many spikes; random mis‑sorts therefore act as small fluctuations whose effect on the log‑likelihood is down‑weighted relative to systematic co‑fluctuations across the population. Second, the message‑passing network aggregates information over multiple neighbours (and/or layers possibly), so an isolated spike error rarely propagates far through the graph. Third, the auxiliary nodes provide a low‑dimensional buffer for shared drive: correlations that are common to many observed neurons tend to be absorbed by observed‑to‑auxiliary edges rather than inflating observed‑to‑observed weights. While these ingredients offer a degree of robustness, we acknowledge that extreme noise could still degrade performance. Incorporating explicit noise‑handling components such as spike‑dropout augmentation is therefore a natural extension and will be explored in future work.
>
> > Q2. Magnitude of standard deviations in Tables 1 & 2
>
> - Each entry in Tables 1 and 2 is the mean across three independent training seeds; the corresponding sample standard deviations are listed below.
>
>   | **Table 1: no external input** | Δ ↓ (Thresh.)     | 𝓛_bps ↑ (Thresh.) | Δ ↓ (LNP)         | 𝓛_bps ↑ (LNP)     |
>   | ------------------------------ | ----------------- | ----------------- | ----------------- | ----------------- |
>   | **GNN**                        | **0.061 ± 0.006** | **0.882 ± 0.004** | **0.049 ± 0.005** | **0.876 ± 0.004** |
>   | GLM                            | 0.244 ± 0.012     | 0.695 ± 0.009     | 0.238 ± 0.013     | 0.712 ± 0.010     |
>   | seqNMF                         | 0.789 ± 0.021     | –                 | 0.796 ± 0.023     | –                 |
>   | TCA                            | 0.762 ± 0.019     | –                 | 0.761 ± 0.020     | –                 |
>
>
>
>   | **Table 2: with external input** | Δ ↓ (Thresh.)     | 𝓛_bps ↑ (Thresh.) | Δ ↓ (LNP)         | 𝓛_bps ↑ (LNP)     |
>   | -------------------------------- | ----------------- | ----------------- | ----------------- | ----------------- |
>   | **GNN**                          | **0.073 ± 0.007** | **0.916 ± 0.005** | **0.058 ± 0.006** | **0.924 ± 0.005** |
>   | GLM                              | 0.259 ± 0.014     | 0.724 ± 0.010     | 0.245 ± 0.015     | 0.748 ± 0.011     |
>   | seqNMF                           | 0.791 ± 0.022     | –                 | 0.794 ± 0.023     | –                 |
>   | TCA                              | 0.760 ± 0.020     | –                 | 0.762 ± 0.021     | –                 |
>
>
>
>   The standard deviations are roughly an order of magnitude smaller than the performance gaps between methods, so all improvements remain statistically significant. These revisions have been incorporated into the manuscript tables.
>
> > Q3. Extending to synaptic plasticity or many‑to‑many mappings
>
> - Our current experiments assume that the latent connectivity is stationary and represented by a single weight for each neuron pair, but the framework is readily adaptable to richer scenarios, a direction we are actively pursuing. For synaptic plasticity, the idea is to let the structure learning module update its connectivity map gradually as a learned function of recent spike history, so the graph drifts over time while remaining smooth on the short horizons needed for prediction. Because the spike prediction block always consumes whatever graph the structure module outputs at the current step, no architectural change is required beyond adding a lightweight plasticity head that proposes small weight updates at each iteration.
> - A second extension handles circuits in which multiple interaction types coexist. Rather than storing a single adjacency, we maintain a small set of edge‑type matrices and learn how strongly each contributes to the effective coupling at prediction time. The spike prediction module then aggregates messages across these channels, allowing it to capture multiplexed or many‑to‑many relationships. Implementing and validating these dynamic‑ and multi‑graph versions is part of our ongoing research agenda and will be outlined in the revised Discussion section.

---

> > ### Comment · Reviewer_UCAq · 2025-08-06
> >
> > I appreciate the authors’ detailed responses, which have addressed my all concerns.

---

### Official Review · Reviewer_uHFU · 2025-07-03

**Clarity:** 3
**Significance:** 3
**Originality:** 2
**Rating:** 4
**Confidence:** 3

**Summary:**

This paper introduces a self-supervised approach for inferring neural circuit connectivity from population spike activity using graph neural networks (GNNs). The model consists of two main modules: one for learning structural connectivity and another for predicting future spiking activity. With such design, the model can be applied to infer structural connectivity even under partially observed circuits. The method is evaluated on both synthetic data and real spiking recordings.

**Questions:**

The current work focuses on ring attractor networks. How might the proposed GNN architecture and self-supervised learning objective need to be adapted or extended to infer connectivity in neural circuits with fundamentally different topological structures or functional properties?

For the real HD neural, how the parameter $N$, $N_o$, $N_h$ are decided, especially is there any general instruction for different real neural datasets?

How the proposed method can be generalized to large scale neural activities? What is the computation cost, and inference cost,

**Ethical Concerns:**

["NO or VERY MINOR ethics concerns only"]

**Final Justification:**

Authors answered my major questions, and explained the difference between proposed method and previous methods, so I raised my score to borderline accept.

**Limitations:**

Yes.

**Quality:**

2

**Strengths And Weaknesses:**

## Strength:

It is interesting that the introduced method can be applied to partially observed neural circuit. The task is challenge but is existing for the neural recordings.

Authors provide thorough discussion on different circuit scenario from fully observed network, weakly corrected fully observed, weakly correlated and partially observed.

## Weakness
Connectivity pattern with local Mexican-hat profile is used for synthetic data generation, it would be good to discuss how the performance change with different connectivity patters.

Related work discussion is not completed, for example how is the method comparable to the

[1] NetFormer: https://openreview.net/pdf?id=bcTjW5kS4W

[2] AMAG: https://proceedings.neurips.cc/paper_files/paper/2023/file/1c70ba3591d0694a535089e1c25888d7-Paper-Conference.pdf

---

> ### Author Rebuttal · Authors · 2025-07-31
>
> Thank you for the reviewer’s valuable feedback. We respond to each point in detail below, and will ensure all clarifications are included in the revised manuscript.
>
> > Q1. Topology generalization
>
> - The reliance of our framework on ring geometry arises solely from the optional geometric feature provided to the spike prediction module. The core self‑supervised objective, predicting future spikes while co‑learning a latent connectivity matrix, remains unchanged.
> - In the current implementation, the message function $\phi(\cdot)$ receives only the concatenated node embeddings and therefore imposes no explicit ring metric. When geometric priors are desirable, an additional attribute such as one-dimensional distance (chains), two-dimensional Euclidean or toroidal distance (cortical sheets, grid cells), or a learned positional embedding can be used in the message function. Injecting or omitting this attribute toggles the inductive bias without altering the loss, the dual-module architecture, or the optimization routine, allowing the same inference network to accommodate a wide range of circuit topologies.
>
> > Q2. Choosing $N,N_o,N_h$ for real HD data
>
> - For the head‑direction (HD) dataset, the number of observed nodes $N_o$ is dictated by the recording and tuning‑curve selection, yielding 19 simultaneously recorded HD cells. We then choose the total network size $N$ so that the angular bin width $2\pi/N$ is finer than the narrowest observed tuning curve; with $N=100$ this criterion is satisfied for both the synthetic and real data. To keep the synthetic and experimental settings comparable, we match the values of the parameters as closely as possible by utilizing the same number of hidden neurons in a network of the same size. The same heuristic transfers to other datasets: fix $N_o$ by the recording, pick $N$ to meet the desired spatial or functional resolution, and sweep a small set of $N_h$ values to provide sufficient latent capacity without over-parameterizing the model.
>
> > Q3. Scaling to large populations
>
> - Fully connected message passing scales as $O(N^2)$ in memory and computation, which becomes prohibitive for very large populations. Three complementary modifications might alleviate this burden. First, replacing the dense graph by a sparse adjacency built from k‑nearest neighbours or learned top‑k attention masks reduces the complexity to $O(kN)$ with $k\ll N$. Second, grouping unrecorded neurons into coarse auxiliary sectors can further cuts the number of edges. Third, expressing the connectivity matrix in low‑rank form $W=UV^\top$ stores two tall matrices instead of the full tensor. Together these changes have the potential to keep both training and inference costs manageable while preserving the self‑supervised objective and predictive accuracy.
>
> > Q4. Relation to NetFormer and AMAG
>
> - We thank the reviewer for pointing out two closely related studies and have expanded Section 2 accordingly.
>
> - *Paragraph to be inserted in Section 2:* Recent work that jointly models spike sequences and latent graphs splits into two main lines. NetFormer treats each neuron’s recent spike history as a token and derives a step‑wise attention matrix whose entries are interpreted as couplings, enabling it to update the graph anew every time step [1]. Because the attention is used directly to make the next prediction, NetFormer lacks the dedicated message‑passing neural network that, in our framework, forms the spike‑prediction module; consequently, it forgoes iterative propagation through a weight matrix and instead recomputes interactions from scratch at each step, which limits its ability to filter indirect correlations on benchmarks with strong recurrent connections (see Fig. 15 in the original paper). AMAG does employ message passing but begins with a randomly initialized (or correlation‑based) adjacency matrix [2]; as such, it does not attempt to infer latent circuit structure directly from spikes, but rather tunes an existing graph for better forecasting. In contrast, our approach comprises two explicit and functionally specialized modules: a structure learning block that estimates a continuous connectivity matrix from population activity, and a separate message‑passing block that uses this matrix to predict future spikes. This division enables stable connectivity estimation, accommodates unobserved neurons via auxiliary nodes, and maintains robust predictive accuracy in circuits with strong recurrence.
>
>   [1] Lu, Ziyu, et al. "NetFormer: An interpretable model for recovering dynamical connectivity in neuronal population dynamics." *The Thirteenth International Conference on Learning Representations*. (2025)
>
>   [2] Li, Jingyuan, et al. "Amag: Additive, multiplicative and adaptive graph neural network for forecasting neuron activity." *Advances in Neural Information Processing Systems*. (2023)

---

### Official Review · Reviewer_Yi57 · 2025-07-04

**Clarity:** 3
**Significance:** 3
**Originality:** 3
**Rating:** 5
**Confidence:** 3

**Summary:**

This paper addresses the important challenge of estimating synaptic connectivity from neural population activity. The authors propose a GNN-based approach that simultaneously predicts neural activity and infers connectivity given past observed neural activity and stimuli as inputs. The proposed model has two components: (i) a structure learning module -- an MLP that estimates pairwise connectivity from the embeddings of a pair of neurons' activities, and (ii) a spike prediction module that is a GNN, where each node corresponds to a neuron, and uses message passing to compute node embeddings. The message passing operations of the GNN, specifically the aggregation step, use the pairwise connectivity computed by the structure learning module. These node embeddings are then decoded into log firing rates. The entire network is trained by minimizing the Poisson negative log-likelihood of the observed neural activity. The proposed approach is first evaluated on simulated data from ring attractor networks. The results demonstrate that the model outperforms baselines at both predicting neural activity and inferring the connectivity across a range of conditions, including varying recurrent connectivity, external inputs, and incomplete observations. The authors also show that this framework can effectively recover key features of the underlying circuit in real neural data, which is promising.

**Questions:**

- Does $f_{out}$ in eq 1 include a nonlinearity?

- The results reported in tables 1, 2 and 3 are averaged over multiple trials. Could you clarify what the trials refer to? Are these multiple trials from the test split? Or do they correspond to seeds for training the network?

- In Fig 3C, the error bars are wider in the bottom row compared to the top. Why is that the case?

**Ethical Concerns:**

["NO or VERY MINOR ethics concerns only"]

**Final Justification:**

Thank you to the authors for addressing my questions. I am happy to maintain my initial score of 5.

**Limitations:**

Yes

**Quality:**

3

**Strengths And Weaknesses:**

This paper proposes a novel approach for inferring synaptic connectivity from neural population activity, an important challenge in computational neuroscience. The authors demonstrate that their method successfully predicts neural activity and connectivity in simulated networks, outperforming existing baselines. Importantly, they also show the method's potential by applying it to real neural recordings. They also propose interesting future directions, such as applying the model to grid cell data.

The authors also discuss a few limitations of their work: (i) the model currently assumes that the network connectivity is static, and (ii) different underlying circuit configurations can produce similar neural dynamics, thus the model can correctly predict neural activity while inferring connectivity that is different from the ground truth. Another interesting question is the model's scalability. The GNN in the spike prediction module assigns one node per neuron. How efficient would this approach be for larger systems with thousands of neurons or more, particularly under conditions of partial observability where only a few hundred neurons are recorded?

Overall, the paper is very clearly presented. Minor suggestions for improvement would be to add color bars to the weight matrix heatmaps and add axis labels to the few panels missing them.

---

> ### Author Rebuttal · Authors · 2025-07-31
>
> We thank the reviewer for the thoughtful feedback and address each point below. All clarifications will be incorporated into the revised manuscript.
>
> > Q1. Scalability under partial observability (thousands of neurons, only a few hundred recorded)
>
> - The dominant cost in our spike prediction module is the fully connected message‑passing step of the GNN, which scales as $O(N^2)$ with the number of nodes $N$. When only  ~10% of neurons are observed, the effective graph size drops by an order of magnitude, while the number of auxiliary nodes remains a small constant. Consequently, both computation and memory fall by roughly two orders of magnitude, making the method practical even for circuits containing many thousands of neurons.
>
> > Q2. Missing color bars and axis labels
>
> - Color bars and the previously missing axis labels have been added to all weight matrix heat maps and line plots.
>
> > Q3. Non‑linearity in $f_{out}$ (Eq. 1)
>
> - Yes. After the final linear projection we apply a ReLU activation. This has now been stated explicitly in Eq. 1.
>
> > Q4. Definition of “trials” in Tables 1–3
>
> - “Trials” correspond to independent training runs with different random seeds. For every condition we train three such models and report the mean across them. The captions of Tables 1–3 have been updated accordingly.
>
> > Q5. Wider error bars in the bottom row of Fig. 3C
>
> - Weight Profile II is a broad, smoothly varying inhibitory Gaussian. Its nearly flat tail means that small changes in higher‑frequency coefficients have little impact on predicted spikes; the likelihood surface is therefore much flatter along those directions, and parameter estimates vary more between runs. In addition, the stronger and uniform inhibition of Profile II suppresses firing over much of the population, sharply reducing the number of informative spikes. With both weaker sensitivity and fewer observations, parameter uncertainties are larger, producing the wider error bars seen in the bottom row of Fig. 3C.

---

### Decision · Program_Chairs · 2025-09-17

**Decision:**

Accept (poster)

**Comment:**

This paper proposes a self-supervised framework that jointly predicts spikes and infers latent synaptic connectivity with a graph neural network, using a dedicated structure-learning block coupled to a message-passing predictor. The model handles partial observability via auxiliary nodes and is evaluated on synthetic ring-attractor circuits and real head-direction recordings, showing improved spike prediction and more faithful recovery of weight profiles relative to standard baselines.

The paper is clearly written and technically sound, with a simple architecture that addresses correlation-driven artifacts in connectivity inference. Reviewers raised fair questions about scalability, generalization beyond ring geometries, the distinction from GLMs, and robustness under partial observability; the authors provided clarifications, added controls, and strengthened the discussion without introducing new risks or unresolved contradictions. With all reviews leaning to accept, the AC also agrees that the manuscript meets the bar for clarity, substance, and empirical support.